# Decoding the biogenesis of HIV-induced CPSF6 puncta and their fusion with nuclear speckles

Chiara Tomasini[1†], Celine Cuche[1†], Selen Ay[1], Maxence Collard[1], Bin Cui[2], Mohammad Rashid[2], Shaoni Bhattacharjee[1], Bruno Tello-Rubio[1], Julian Buchrieser[3], Charlotte Luchsinger[2], Cinzia Bertelli[2], Vladimir Uversky[4], Felipe Diaz-Griffero[2]*, Francesca Di Nunzio[1]*

[1]Institut Pasteur, Advanced Molecular Virology Unit, Department of Virology, Université Paris Cité, Paris, France; [2]Albert Einstein College of Medicine, Department of Immunology and Microbiology, New York, United States; [3]Institut Pasteur, Virus and Immunity Unit, Department of Virology, Université Paris Cité, Paris, France; [4]Department of Molecular Medicine and USF Health Byrd Alzheimer's Research Institute, Morsani College of Medicine, University of South Florida, Tampa, United States

*For correspondence: felipe.diaz-griffero@einsteinmed. edu (FD-G); dinunzio@pasteur.fr (FDN)

[†]These authors contributed equally to this work

Competing interest: The authors declare that no competing interests exist.

## eLife Assessment

This is a **valuable** study that presents **convincing** evidence on the genesis of the CPSF6 condensates that form upon HIV-1 infection and the specific molecular determinants involved in their formation, as well as their interactions with SRRM. The study could be strengthened by assessing the relevance of their findings to infection, and in particular, with reverse transcription and gene expression

**Abstract** Viruses rely on host cellular machinery for replication. After entering the nucleus, the HIV genome accumulates in nuclear niches where it undergoes reverse transcription and integrates into neighbouring chromatin, promoting high transcription rates and new virus progeny. Despite antiretroviral treatment, viral genomes can persist in these nuclear niches and reactivate upon treatment interruption, raising the possibility that they could play a role in the establishment of viral reservoirs. The post-nuclear entry dynamics of HIV remain unclear, and understanding these steps is critical for revealing how viral reservoirs are established. In this study, we elucidate the formation of HIV-induced CPSF6 puncta and the domains of CPSF6 essential for this process. We also explore the roles of nuclear speckle (NS) scaffold factors, SON and SRRM2, in the biogenesis of these puncta. Through genetic manipulation and depletion experiments, we demonstrate the key role of the intrinsically disordered region of SRRM2 in enlarging NSs in the presence of the HIV capsid. We identify the FG domain of CPSF6 as essential for both puncta formation and binding to the viral core, which serves as the scaffold for CPSF6 puncta. While the low-complexity regions modulate CPSF6 binding to the viral capsid, they do not contribute to puncta formation, nor do the disordered mixed charge domains of CPSF6. Interestingly, the FG peptide facilitates viral replication. These results demonstrate how HIV evolved to hijack host nuclear factors, enabling its persistence in the host. Of note, this study provides new insights into the underlying interactions between host factors and viral components, advancing our understanding of HIV nuclear dynamics and offering potential therapeutic targets for preventing viral persistence.

## Introduction

Since the discovery of HIV (*Barré-Sinoussi et al., 1983*), the initial stages of the viral life cycle, such as the reverse transcription (the conversion of the viral RNA in DNA) and the uncoating (loss of the capsid), have been understood to mainly occur within the host cytoplasm. Only the pre-integration complex, carrying the fully reverse-transcribed viral DNA, was believed to enter the nucleus for integration into the host chromatin (*Suzuki and Craigie, 2007*). Recent studies highlighted that the viral genome is transported in the nucleus via a shuttle that shields it from the hostile cellular environment (*Rasaiyaah et al., 2013*). This shuttle is constituted by the viral capsid (*Blanco-Rodriguez and Di Nunzio, 2021*; *Blanco-Rodriguez et al., 2020*; *Chen et al., 2016*; *Selyutina et al., 2020*; *Yamashita and Emerman, 2004*; *Zila et al., 2021*), which comprises 250 hexamers (depending on the size and shape of the capsid) and 12 pentamers (*Pornillos et al., 2009*). Within the structure of the capsid, hydrophobic pockets exist between hexamers, which serve as targets for various nucleoporins, especially the ones carrying FG repeats, facilitating the translocation of the capsid through the nuclear pore complex (NPC) (*Buffone et al., 2018*; *Di Nunzio, 2013*; *Di Nunzio et al., 2012*; *Lelek et al., 2015*; *Matreyek et al., 2013*; *Price et al., 2014*). Recent studies suggest that HIV uses multiple FG regions of several nucleoporins to translocate through the NPC, acting as a chaperone by itself (*Dickson et al., 2024*; *Fu et al., 2024*). Using a reductionist system of Nup98 condensates, it has been demonstrated that FG-mediated phase partitioning identifies specific sites on the capsid that allow it to interact autonomously with these phases. These findings complement the evolving understanding of the early stages of HIV infection, which has been revisited in recent years. They unveil that crucial stages of early viral infection occur within the host nucleus (*Burdick et al., 2020*; *Dharan et al., 2020*; *Francis et al., 2020*; *Scoca et al., 2023*; *Selyutina et al., 2020*). Notably, it has been shown that the pre-integration complex forms within the host nucleus (*Müller et al., 2021*; *Scoca et al., 2023*), and incoming viral RNA genomes accumulate in nuclear niches containing cleavage and polyadenylation specificity factor subunit 6 (CPSF6), RNA-binding protein SON and splicing component, 35 kDa SC35, also known as serine/arginine-rich splicing factor 2 (SRSF2) (*Francis et al., 2020*; *Rensen et al., 2021*; *Scoca et al., 2023*). However, the mechanistic requirements that govern post-nuclear entry phases, which are crucial for a successful viral infection and the establishment of viral reservoirs, remain enigmatic.

Specifically, the mechanism underlying the formation of nuclear niches containing viral components and nuclear speckle (NS) factors, such as SC35—traditionally used as a marker for NS, remains unclear. However, recent studies have revealed that SON and serine/arginine repetitive matrix protein 2 (SRRM2) are essential for NS biogenesis (*Fu and Maniatis, 1990*). Particularly, intrinsically disordered regions (IDRs) play a critical role in the NS formation (*Ilik et al., 2020*). These membraneless organelles (MLOs) fulfil various cellular functions besides splicing. Recent evidence indicates a direct role of NSs in cellular transcription regulation, as their spatial proximity correlates with gene expression amplification, as demonstrated by live-cell imaging of heat-shock responsive genes (*Chen et al., 2018*; *Zhang et al., 2021*). HIV, being a virus capable of generating new particles through splicing and integrating into active host genes, finds NSs highly conducive for viral replication. HIV particles, along with their RNA genome, accumulate within nuclear MLOs enriched in NS factors (*Ay et al., 2025*; *Rensen et al., 2021*; *Scoca et al., 2023*). Notably, CPSF6, a paraspeckle factor first identified as a viral partner by KewalRamani's laboratory (*Lee et al., 2010*), has been clearly detected in HIV-induced CPSF6 puncta (*Ay et al., 2025*; *Francis et al., 2020*; *Lee et al., 2010*; *Luchsinger et al., 2023*; *Rensen et al., 2021*; *Scoca et al., 2023*). These puncta serve as hubs for nuclear reverse transcription and the formation of pre-integration complexes, which generate active proviruses detected outside but in close proximity to NSs (*Ay et al., 2025*; *Li et al., 2021*; *Scoca et al., 2023*).

In our study, we aim to elucidate how HIV-induced CPSF6 puncta form and identify the NS factors involved in their formation. CPSF6, along with NS factors, contains IDRs that can guide HIV-1 to the correct nuclear location for successful infection or allow the virus to remain sequestered during drug treatment, forming reservoirs. Importantly, viral reservoirs are the major bottleneck for curing the infection.

In this study, we investigate which disordered domain of CPSF6 is responsible for the recruitment of the viral core and generation of HIV-induced CPSF6 puncta in the host nucleus. Simultaneously, we elucidate the key component of NSs that, through its IDRs, enables fusion with HIV-induced CPSF6 puncta, likely stabilizing them. Notably, we observed that the virus rebounds when anti-reverse

transcription drugs are removed, but only if nuclear niches containing HIV, NS factors, and CPSF6 are present. If these niches are pharmacologically dismantled, viral rebound does not occur.

Overall, studying the biogenesis of HIV-induced nuclear niches is crucial for understanding how the virus navigates and persists in the nucleus of infected cells, and for designing new antiretroviral strategies.

## Results

### Critical role of HIV-induced CPSF6 puncta in restoring nuclear reverse transcription after anti-reverse transcriptase (RT) therapy discontinuation

Upon nuclear entry, HIV enhances the formation of CPSF6 puncta, where reverse transcription ends. The treatment with the reversible RT inhibitor nevirapine (NVP) can trap the viral RNA genome in these nuclear niches (*Ay et al., 2025*; *Rensen et al., 2021*; *Scoca et al., 2023*). Once NVP is removed, the trapped vRNA can resume RT entirely within the nucleus, a process we term nuclear reverse transcription. Here we demonstrate that this phenomenon depends on the presence of CPSF6 puncta, which are mainly composed of CPSF6 proteins and viral cores. Disruption of these puncta by high doses of PF74 (25 µM) significantly impairs the rebound of the RT, as evidenced by the absence of luciferase expression, similar to what is observed with complete NVP treatment (*Figure 1A, B*).

### CPSF6 FG domain is required for HIV-induced puncta formation

Formation of CPSF6 puncta upon HIV-1 infection hinges on two key events: the entry of the HIV-1 core into the nucleus and the binding of CPSF6 to the HIV-1 core (*Blanco-Rodriguez and Di Nunzio, 2021*; *Blanco-Rodriguez et al., 2020*; *Buffone et al., 2018*; *Zila et al., 2021*). To determine the contribution of CPSF6's disordered domains to the formation of CPSF6 puncta upon HIV-1 infection, we correlated the binding of CPSF6 to the HIV-1 core with the formation of CPSF6 puncta. To this end, we first generated CPSF6 knockout (KO) THP-1 cells (*Figure 1C, D*) to eliminate the interference from the endogenous protein, which could affect the interpretation of results regarding the role of the analysed CPSF6 domains. CPSF6 depletion in THP-1 cells was performed by CRISPR–Cas9 technology. To completely eliminate the expression of the CPSF6 gene, we selected single clones by limiting dilution. We identified a clone that was completely KO for CPSF6, confirmed through western blot and immunofluorescence (*Figure 1C, D*), and we infected this clone and the control clone with HIV. CPSF6 puncta were detected only in the control-infected cells and not in the KO clone (*Figure 1D*; *Figure 1—figure supplement 1*). The viral integrase (IN) was observed within CPSF6 puncta, consistent with previous studies (*Francis et al., 2020*; *Rensen et al., 2021*; *Scoca et al., 2023*), but absent in CPSF6 KO cells where viral IN was observed in the cytoplasm (*Figure 1D*; *Figure 1—figure supplement 1*). Thus, we used KO cells for CPSF6 to assess the role of selected CPSF6 domains in HIV-induced condensates (*Figure 1—figure supplement 2*). We designed various CPSF6 deletion mutants (*Figure 2A*) to specifically assess the significance of the main disordered regions of CPSF6 protein (*Figure 2—figure supplement 1*) such as the FG (phenylalanine and glycine) motif, the low complexity regions (LCRs), and the mixed charge domain (MCD), in the ability of CPSF6 to bind to the core and facilitate the formation of CPSF6 puncta. We investigated the role of the FG peptide by generating a mutant that exclusively lacks the FG peptide (ΔFG). Previous in vitro studies have shown that the FG peptide binds to the hydrophobic pocket formed between capsid hexamers (*Buffone et al., 2018*; *Price et al., 2014*). Here, we want to investigate the role of FG peptide in the context of the protein during the viral infection.

To further explore this, we developed an alternative plasmid by expanding the FG peptide deletion to include surrounding prion-like LCRs (ΔFG ΔLCR). These regions, outside the CPSF6 context, have been identified as crucial for facilitating strong CPSF6 binding to capsid lattices (*Wei et al., 2022*). In our study, we aim to evaluate their role within a more physiological setting. Additionally, we assessed a CPSF6 variant that carries the 15-mer FG peptide flanked by non-LCR sequences, such as those derived from Beta-adducin (ADD2), kindly provided by Mamuka Kvaratskhelia (ΔLCR + ADD2). These protein segments are known for their high flexibility, akin to the LCR of CPSF6. Furthermore, to elucidate the contribution of the LCRs of CPSF6 in the formation of CPSF6 puncta, we generated a mutant lacking both LCRs (ΔLCR). Analysis of the MCD contribution to both the ability of CPSF6 to bind to

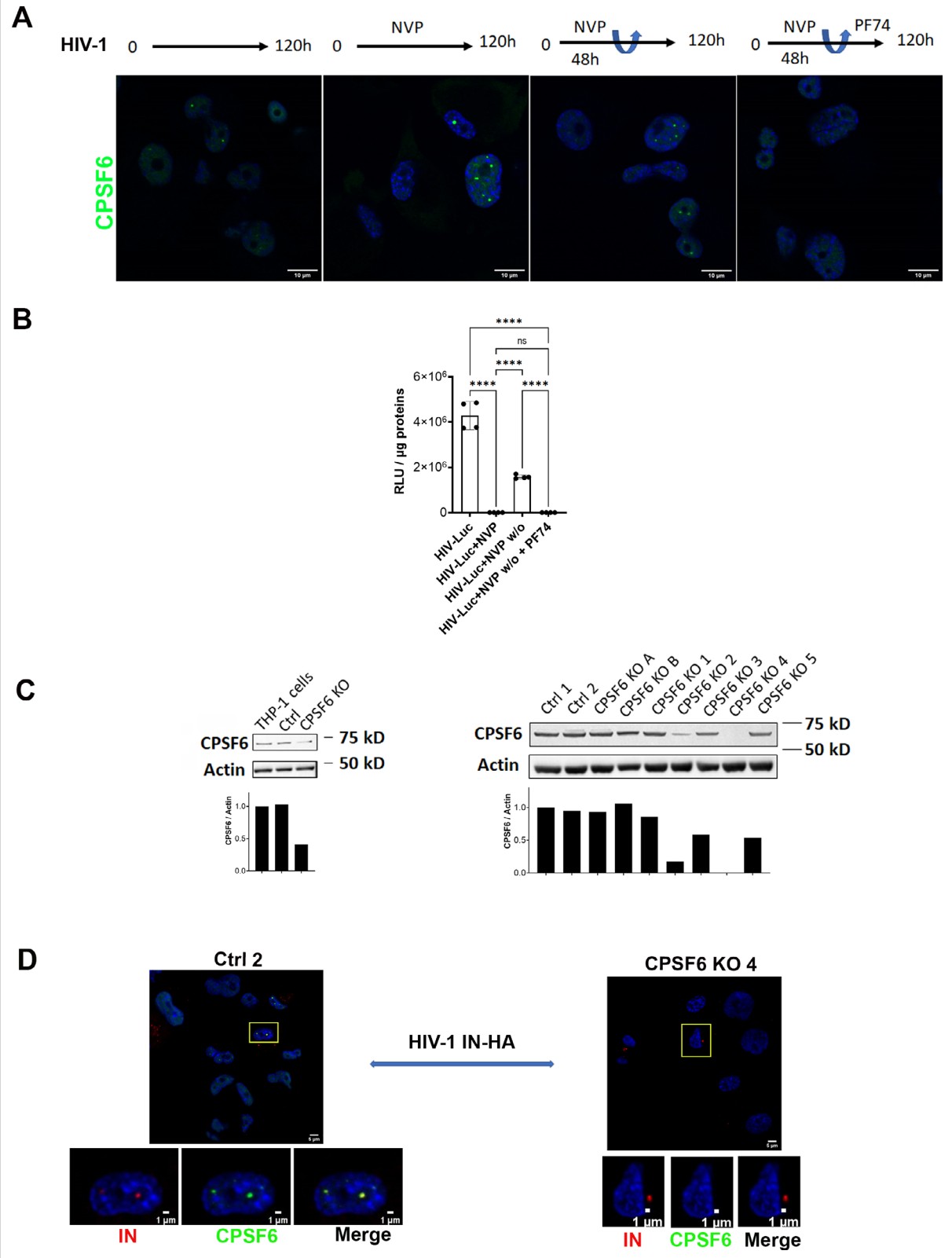

**Figure 1.** Role of HIV-induced CPSF6 puncta in the nuclear reverse transcription upon removal of NVP. (**A**) THP-1 cells, infected with VSV-G Δenv HIV-1 (NL4.3) ΔR LUC (MOI 10) in presence or not of Nevirapine (10 µM) for 5 days, or in presence of Nevirapine (10 µM) for 2 days and then the remaining 3 days without drug or in presence of Nevirapine (10 µM) for 2 days then in presence of PF74 (25 µM). Confocal microscopy images, to verify the presence of CPSF6 puncta, the cells are stained with anti-CPSF6 antibody (green). Nuclei are stained with Hoechst (blue). Scale bar 10 µm. (**B**) Luciferase

*Figure 1 continued on next page*

*Figure 1 continued*

assay, to verify luciferase expression in the aforementioned samples. Luciferase values were normalized by total proteins revealed with the Bradford kit. One-way ANOVA statistical test with multiple comparison was performed (****$p < 0.0001$; *$p < 0.05$; ns, $p > 0.05$). Data are representative of two independent experiments. (**C**) Western blots demonstrate CPSF6 depletion using a specific antibody against CPSF6 in THP-1 cells subjected to CRISPR–Cas9 methods: CRISPR–Cas9 bulk (left), and CRISPR–Cas9 clones selected by limiting dilution (right). Each condition is normalized for actin labelling. The ratio between the intensity signal of CPSF6 and actin was analysed via ImageJ and is plotted below each western blot. (**D**) Confocal microscopy images of THP-1 ctrl CRISPR clone 2 cells (Ctrl 2) and THP-1 duplex1-2-3 CRISPR clone 4 cells (CPSF6 KO 4) infected with VSV-G/HIV-1ΔEnvIN$_{HA}$ LAI (BRU) -vpx (MOI 10) in the presence of Nevirapine (10 μM). The cells are stained 30 hr p.i. with anti-CPSF6 antibody and anti-HA antibodies to detect HA tagged integrase (IN).

The online version of this article includes the following source data and figure supplement(s) for figure 1:

**Source data 1.**

**Source data 2.**

**Source data 3.**

**Source data 4.**

**Source data 5.**

**Source data 6.**

**Figure supplement 1.** Multiple examples of confocal microscopy images of THP-1 ctrl CRISPR clone 2 cells and THP-1 CPSF6 KO 4 cells infected with VSV-G/HIV-1ΔEnvIN$_{HA}$ LAI (BRU) -vpx (MOI 10) in the presence of Nevirapine (10 μM).

**Figure supplement 2.** Multiple examples of THP-1 CPSF6 KO clone 4 cells transduced with different LVs carrying CPSF6 WT or mutants and stained with CPSF6 and HA antibody 30 hr p.i.

the core and formation of CPSF6 puncta was achieved by deleting the MCD and adding 3 nuclear localization signals (3xNLS ΔMCD) since the deletion of the MCD results in a protein that localizes mainly into the cytoplasm (*Figure 2A, B*; *Figure 2—figure supplement 2*).

To correlate the ability of CPSF6 to bind to the HIV-1 core with formation of CPSF6 puncta, we expressed wild-type and mutant CPSF6 constructs in THP-1 cells knockout for CPSF6. Subsequently, we infected these cells with HIV-1 and analysed the presence or absence of CPSF6 puncta at 24 hr post-infection. Importantly, for the imaging experiment, we expressed CPSF6 WT and mutants without tags to avoid the formation of aggregates that could interfere with our conclusions. Our data show that HIV-induced CPSF6 puncta can form extremely rarely with the deletion mutant CPSF6 ADD2ΔLCR and with the mutant lacking the FG or both the FG peptide and the LCRs (*Figure 2B–D*; *Figure 1—figure supplement 2*). However, when we analysed the role of the MCD domain in CPSF6 puncta formation, which was indicated to be important for condensing CPSF6 in NS (*Greig et al., 2020*), comparing the number of CPSF6 WT puncta induced by HIV infection with CPSF6 mutants revealed that the MCD domain does not play a critical role in HIV-induced CPSF6 puncta formation (*Figure 2B–D*). In addition, we observed that the majority of analysed CPSF6 3xNLSΔMCD puncta contain vRNA inside, similar to CPSF6 WT puncta (*Figure 2E*), thus corroborating the lack of a role for this intrinsically disordered domain in HIV-induced CPSF6 puncta. Since the NLS domain from SV40, which replaces the MCD, is highly basic and could potentially induce condensates, we fused CPSF6 with a non-basic NLS (PY-NLS) or removed the NLS entirely. Even though these two proteins do not efficiently enter the nucleus, the few that do manage to reach the nucleus can host viral particles, as evidenced by the presence of IN. Many viruses are typically blocked in the cytoplasm due to the presence of these mutants that are mainly cytoplasmic. However, because we used a high viral dose, the blockage in the cytoplasm was not complete. As a result, the viruses that successfully entered the nucleus induced the formation of puncta associated with CPSF6-deleted mutants, indicating that the MCD is not critical for the formation of HIV-induced CPSF6 puncta (*Figure 2F*). Similar to the MCD, when we compared CPSF6 truncated for the LCRs with CPSF6 WT, we observed that the LCRs do not contribute to CPSF6 puncta formation. Therefore, the FG peptide alone, without the LCRs, is the only CPSF6 domain required for their formation (*Figure 2B–F*; *Figure 1—figure supplement 2*).

Next, we tested the ability of the different CPSF6 deletion mutants for their ability to bind the viral core using a previously described capsid binding assay (*Selyutina et al., 2018*). Wild-type and mutant CPSF6 proteins were expressed in human 293T cells at similar levels (INPUT) (*Figure 3A*). Extracts containing wild-type and mutant CPSF6 proteins were incubated with stabilized HIV-1 capsid tubes for 1 hr at 25°C in the presence of 10 μM of PF74, which is a small molecule that competes with CPSF6 for

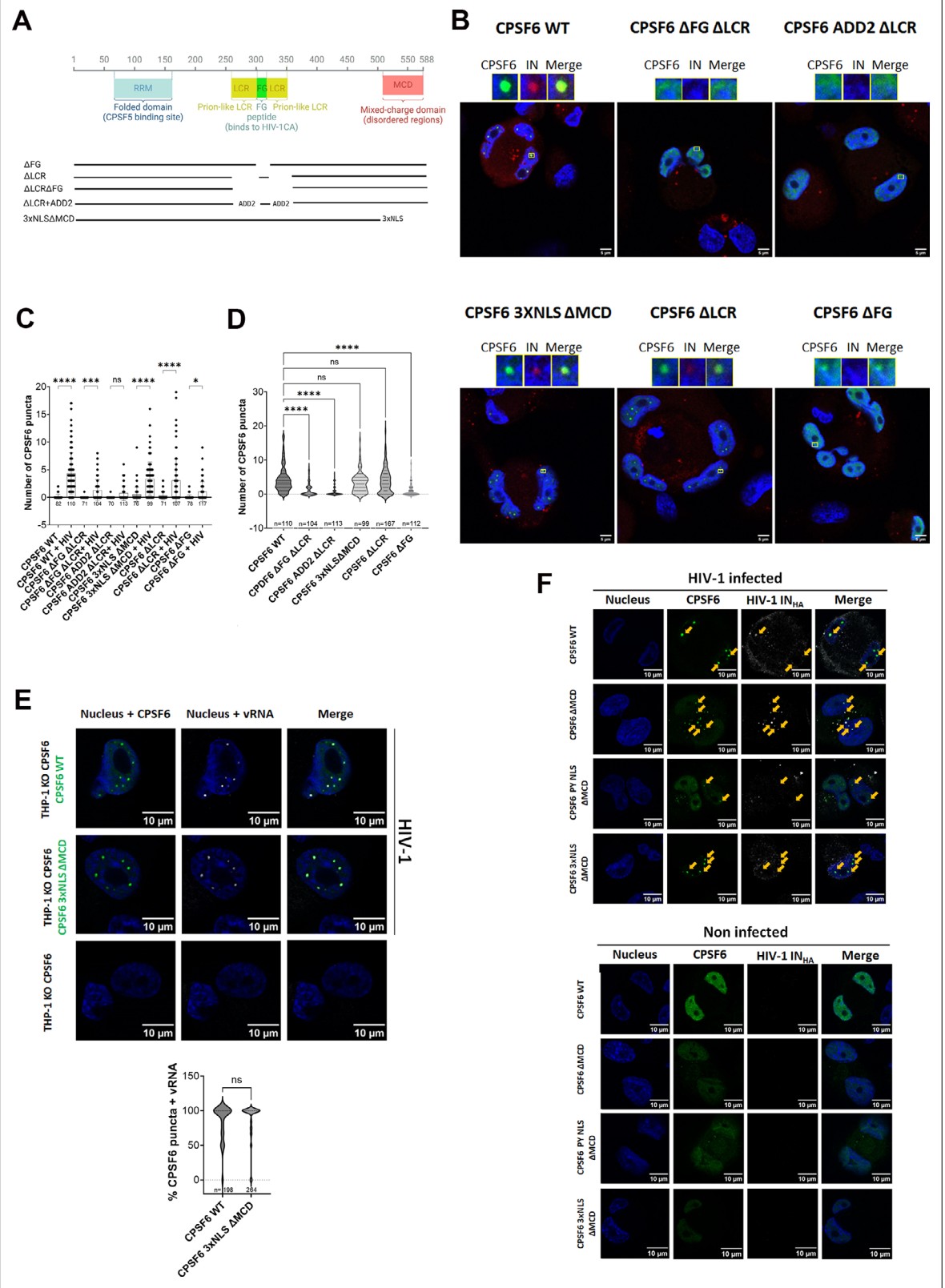

**Figure 2.** Role of CPSF6 domains in HIV-induced CPSF6 puncta. (**A**) Schema of CPSF6 isoform 588 aa deletion mutants. (**B**) Confocal microscopy images of THP-1 CPSF6 KO cells, transduced with different mutants of CPSF6, infected with VSV-G/HIV-1ΔEnvIN_HA LAI (BRU) -vpx (MOI 10) in presence of Nevirapine (10 μM). The cells are stained with CPSF6 and HA antibody 30 hr p.i. Scale bar 5 μm. (**C**) Analysis of the number of CPSF6 puncta in THP-1 CPSF6 KO cells transduced with different mutants of CPSF6, not infected or infected in the presence of Nevirapine (10 μM) (the number of analysed

*Figure 2 continued on next page*

*Figure 2 continued*

cells is shown under the *x*-axis). Statistical test: ordinary one-way ANOVA (****p < 0.0001; ***p < 0.001; *p < 0.05; ns, p > 0.05). (**D**) The plot compares the number of CPSF6 puncta per cell in THP-1 CPSF6 KO cells transduced with different mutants of CPSF6, infected with HIV-1 in the presence of Nevirapine (10 μM). Statistical test: ordinary one-way ANOVA (****p < 0.0001; ns, p > 0.05). (**E**) Confocal microscopy images of THP-1 CPSF6 KO clone 4, non-transduced and non-infected or transduced with WT CPSF6 and CPSF6 3xNLSΔMCD and infected with VSV-G/HIV-1ΔEnvIN$_{HA}$ LAI (BRU) -vpx (MOI 10) in presence of Nevirapine (10 μM). Immuno-RNA FISH: the cells are stained with CPSF6 (green) antibody and with 24 probes against HIV-1 Pol sequence (grey) (RNA-FISH) 25 hr p.i. Nuclei are stained with Hoechst (blue). Scale bar 10 μm. Violin plot presenting the percentage of CPSF6 puncta colocalizing with the viral RNA in THP-1 CPSF6 KO clone 4 cells transduced with LVs expressing CPSF6 WT or CPSF6 3xNLSΔMCD (respectively, n = 73 and n = 103) and infected with VSV-G/HIV-1ΔEnvIN$_{HA}$ LAI (BRU) -vpx (MOI 10) in presence of Nevirapine (10 μM). A total of 198 CPSF6 WT puncta and 264 CPSF6 3xNLSΔMCD puncta were counted. Statistical test: unpaired *t*-test, ns, p > 0.05. (**F**) Confocal microscopy images of THP-1 KO CPSF6 cells transduced with WT CPSF6 and CPSF6 ΔMCD without NLS, with 3xNLS or with PY NLS, respectively. Cells were differentiated for 3 days, transduced with CPSF6 lentiviral vectors (MOI 1) for 3 days and infected for 24 hr with VSV-G/HIV-1ΔEnvIN$_{HA}$ LAI (BRU) -vpx (MOI 10) in the presence of Nevirapine (10 μM). The panels show transduced and uninfected cells. CPSF6 and the IN tagged with the HA are labelled with anti-CPSF6 (green) and anti-HA (white) antibodies, respectively. Nuclei are stained with Hoechst (blue). The arrows show CPSF6 puncta in colocalization with IN-HA. Scale bar 10 μm.

The online version of this article includes the following source data and figure supplement(s) for figure 2:

**Source data 1.**

**Source data 2.**

**Source data 3.**

**Source data 4.**

**Source data 5.**

**Source data 6.**

**Figure supplement 1.** Per-residue intrinsic disorder propensity of the CPSF6 isoform 588 aa evaluated by the Rapid Intrinsic Disorder Analysis Online platform (RIDAO) (*Dayhoff and Uversky, 2022*) that yields results for IU-Pred_short (yellow line), IUPred_long (blue line), PONDR VL3 (green line), PONDR VLXT (black line), PONDR VSL2 (red line), and PONDR FIT (pink line) and computes a mean disorder score for each residue based on these predictors (MDP, thick, dark pink, dashed line).

**Figure supplement 2.** Lentiviral vector transduction of phorbol 12-myristate 13-acetate (PMA)-differentiated THP-1 cells expressing CPSF6 ΔMCD fused to mNeonGreen (left), CPSF6 NLSΔMCD fused to mNeonGreen (centre), and CPSF6 3xNLSΔMCD fused to mNeonGreen (right).

binding to the hydrophobic pocket formed between hexamers that constitute the viral core (*Buffone et al., 2018*; *Price et al., 2014*). HIV-1 capsid stabilized tubes were washed, and the bound proteins were eluted using Laemli buffer (BOUND). For every construct, the percentage of bound protein relative to input in the presence or absence of PF74 is shown (*Figure 3B*). Our results revealed that the absence of the FG peptide (ΔFG) entirely abolished CPSF6's ability to bind to the viral core. In agreement, simultaneous deletion of the FG motif and LCRs (ΔFG ΔLCR) resulted in a construct unable to bind to the viral core. Similar outcomes were observed when the LCRs were replaced with sequences derived from ADD2, even if the FG was present.

LCR-FG is notably more disordered than ADD2-FG, containing a high proportion of prolines (48 out of 98 residues), which makes it mostly non-foldable (*Figure 4A–H*). Since proline is a structure-disrupting residue, LRC-FG is not expected to adopt any secondary structure. In contrast, ADD2-FG contains fewer prolines (15 out of 98 residues) but has many charged residues. It is predicted to form two short α helices and a β strand, arranged as: α helix–FG–β strand–α helix (*Figure 4E*). ADD2-FG may form a flexible collapsed state, as its oppositely charged residues are evenly distributed, potentially allowing polyelectrostatic compaction. This suggests that FG within ADD2-FG may be less accessible for the interaction with the viral core's hydrophobic pocket due to its involvement in this collapsed conformational ensemble (*Figure 4A–E*, *Figure 4—figure supplement 1*). This aligns with the inability of CPSF6 carrying ADD2 in place of the LCRs to induce CPSF6 puncta (*Figure 2B*). On the other hand, the deletion of only the two LCRs, while keeping the FG peptide intact, resulted in unexpected findings. The ΔLCR mutant exhibited a stronger binding affinity for the viral core when compared to the wild-type protein (*Figure 3B*). These results suggest that the LCRs surrounding the FG motif are modulating the affinity of CPSF6 to the viral core, which might be important for function. By contrast, deletion of the MCD (ΔMCD) but retention of other regions, such as the FG peptide and the LCRs, demonstrated a binding affinity to the viral core similar to that of the wild-type protein. These results suggest that the MCD domain is not involved in the binding of CPSF6 to the viral core, which is not surprising since the CPSF6 (1–358), which does not have an MCD, binds to the viral core (*Lee et al., 2010*).

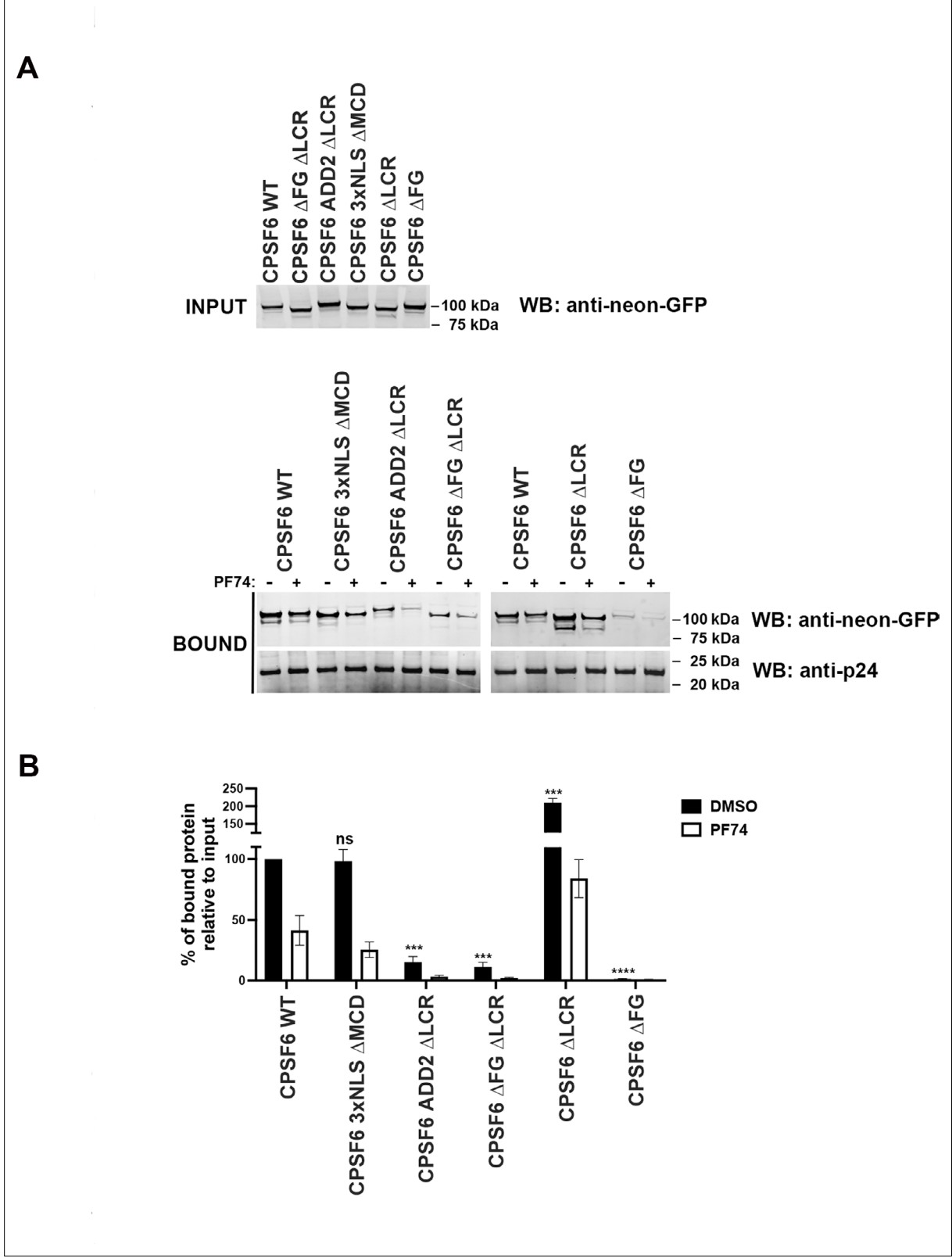

**Figure 3.** Evaluation of CPSF6 deletion mutants' binding capacity to the viral core. (**A**) Ability of wild-type and mutant CPSF6 proteins to bind to the HIV-1 core. Cellular extracts derived from human 293T cells expressing similar levels of the indicated CPSF6 proteins (INPUT) were incubated with HIV-1 capsid stabilized tubes for 1 hr at room temperatures in the presence and absence of 10 µM PF74, as described in materials and methods. As a carrier control, we utilized DMSO. Subsequently, HIV-1 capsid stabilized tubes were washed, and the bound proteins were eluted 1X Laemmli buffer 1X. The

*Figure 3 continued on next page*

*Figure 3 continued*

BOUND fractions were analysed by western blotting using antibodies against neon-GFP and the HIV-1 capsid. (**B**) Experiments were repeated at least three times and the average BOUND fraction relative to the INPUT fraction normalized to wild-type binding is shown for the different CPSF6 mutants. *** indicates a p-value <0.001; **** indicates a p-value <0.0001; and ns indicates no significant difference as determined by unpaired *t*-tests.

The online version of this article includes the following source data for figure 3:

**Source data 1.**

**Source data 2.**

**Source data 3.**

Thus, the viral capsid, through the FG peptide of CPSF6, constitutes the scaffold of HIV-induced CPSF6 puncta.

In summary, our results suggest that the FG peptide is the main determinant involved in the binding of CPSF6 to the viral capsid. Interestingly, our work implies that the LCRs may be modulating the affinity of the FG motif for the viral core (*Figure 3B*). Recognition motifs that mediate protein-protein interactions, such as the FG motif of CPSF6, are usually embedded within longer IDRs that can modulate affinity of the interaction (*Karlsson et al., 2022*). Taken together, our data show that the FG peptide coordinates both the binding to the viral core and the induction of CPSF6 puncta. This coordination suggests that the FG peptide plays a critical dual role in recognizing the viral capsid and facilitating the cellular clustering of CPSF6, which may be part of the cellular response to viral entry.

## Role of FG domain in viral replication

We aimed to investigate the role of the FG domain of CPSF6 in viral infection. To correctly perform the infectivity assay, we generated stable cell clones expressing protein levels comparable to WT cells (*Figure 5A, B*). Importantly, we assessed the reproducibility of our experiments by freezing and thawing these clones. Thus, we performed infectivity assays in THP-1 macrophage-like cells WT, KO for CPSF6 or complemented with CPSF6ΔFG, using both a single-round virus and a replication-competent virus. The results, shown in *Figure 5C, D*, indicate that complete depletion of CPSF6 reduces infectivity, as measured by luciferase expression in a single-round infection (KO: ~65%; ΔFG: ~74%; compared to WT: 100% on average). Notably, a more pronounced defect in viral particle production was observed when WT virus was used for infection (KO: ~21%; ΔFG: ~16%; compared to WT: 100% on average). According to our results, depletion of full-length CPSF6, or expression of a CPSF6 variant lacking only the FG domain, both reduce viral infection compared to cells expressing full-length CPSF6, indicating that the FG domain plays an important role in HIV-1 infection.

## HIV-induced CPSF6 mutants puncta and NSs

The NS factor SC35, commonly used as a marker of NS, has been detected in HIV-induced CPSF6 puncta (*Figure 6A*). In this study, we investigated whether HIV-induced CPSF6 mutant puncta are associated with SC35. We infected cells expressing CPSF6 WT, CPSF6 3xNLS ΔMCD, CPSF6 ΔLCR, CPSF6 ΔMCD, and CPSF6 PY NLS ΔMCD, and examined whether the nuclear puncta formed by the various CPSF6 proteins associate with SC35. Imaging analysis revealed no significant difference in the association of SC35 with CPSF6 WT or the CPSF6 mutants (*Figure 6B*), confirming that the disordered domains, LCRs and MCDs, are dispensable for the formation of HIV-induced CPSF6 puncta that localize in the canonical nuclear niches marked by SC35.

## Biogenesis of HIV-induced CPSF6 puncta carrying NS factors

NS factors, particularly those involved in their biogenesis, such as SON and SRRM2 (*Ilik et al., 2020*), have been identified as constituents of HIV-induced CPSF6 puncta. However, the specific role of NSs in these puncta remains unclear. Most of the existing results have been obtained through immuno-fluorescence experiments conducted several days post-infection. In this study, we conducted a time course experiment to investigate the biogenesis of HIV-induced CPSF6 puncta, which contain NS factors. Our goal was to capture the fusion event between the HIV-induced CPSF6 puncta and NS, providing insights into the dynamics of how HIV manipulates host nuclear structures during infection. We hypothesized that NS factors might either be recruited during the initial formation of HIV-induced CPSF6 puncta, shortly after the virus is released from the nuclear basket of the NPC, or later via the

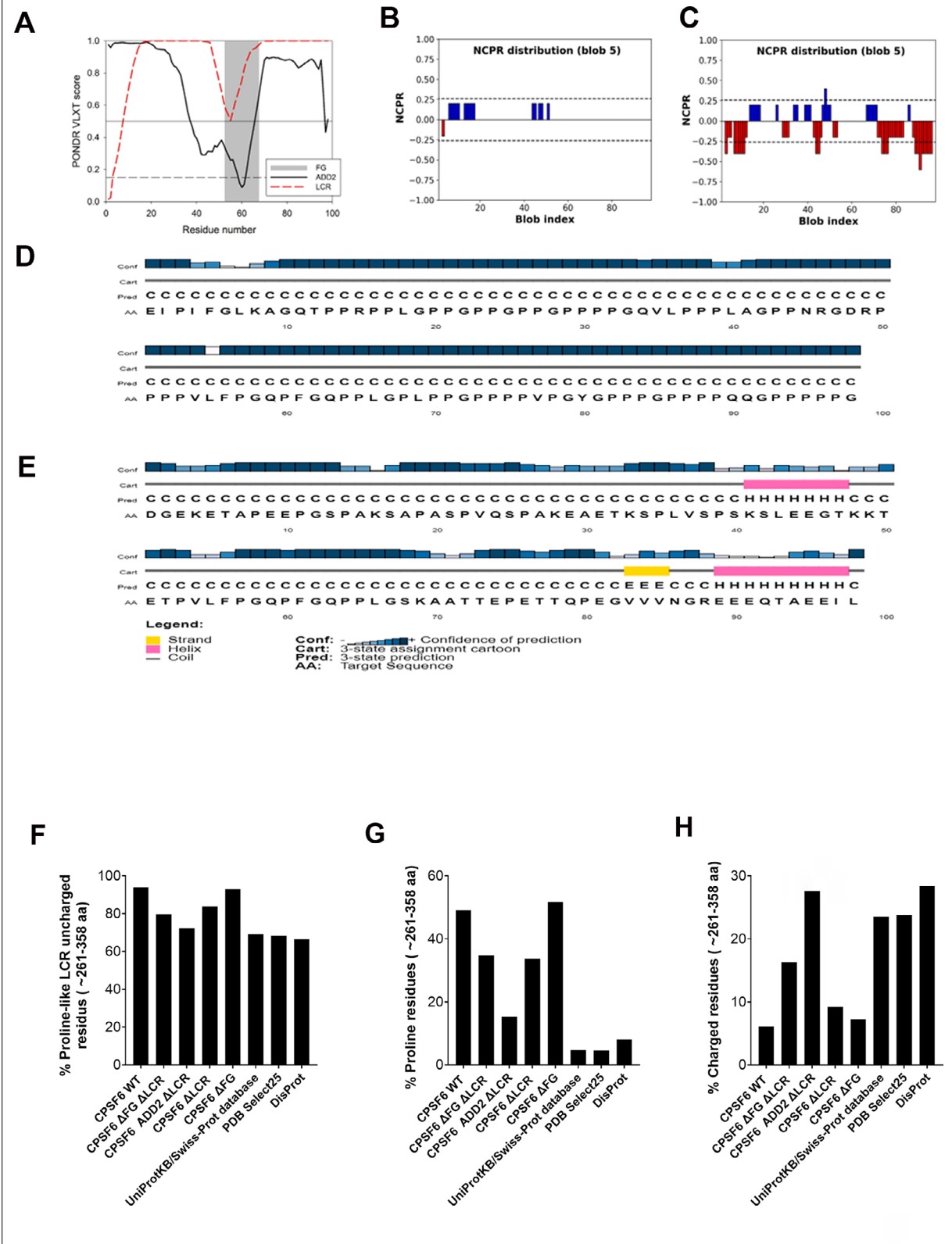

**Figure 4.** Comparison of second structures of ADD2 and low-complexity region (LCR). (**A**) Physicochemical characteristics of the LCR-FG and ADD2-FG sequences. Intrinsic disorder predispositions evaluated by PONDR VLXT. Position of the FR segment within the LCR-FG and ADD2-FG sequences is shown as grey shaded area. (**B**) Linear distribution of the net charge per residue (NCPR) within the LCR-FG sequence evaluated by CIDER. (**C**) Linear distribution of the NCPR within the ADD2-FG sequence evaluated by CIDER. (**D**) Secondary structure propensity of the LCR-FG sequence evaluated

*Figure 4 continued on next page*

*Figure 4 continued*

by PSIPRED. (**E**) Secondary structure propensity of the ADD2-FG sequence evaluated by PSIPRED. (**F**) Analysis of the peculiarities of the amino acid compositions of the intrinsically disordered C-terminal domain (residues 261–358) of human CPSF6 and its different mutants. Relative abundance of prion-like LCR defining uncharged residues in analysed protein segments. (**G**) Relative abundance of proline residues in analysed protein segments. (**H**) Relative abundance of charged residues in analysed protein segments. The values were calculated by dividing numbers of prion-like LCR defining uncharged (Ala, Gly, Val, Phe, Tyr, Leu, Ile, Ser, Thr, Pro, Asn, Gln, Pro) and charged (Asp, Glu, Lys, Arg) residues by the total number of amino acids in the respective protein fragments. Corresponding values for all protein sequences deposited in the UniProtKB/Swiss-Prot database, PDB Select25, and DisProt are shown for comparison.

The online version of this article includes the following source data and figure supplement(s) for figure 4:

**Source data 1.**

**Figure supplement 1.** Sequences of FG and low-complexity regions (LCRs) or substituted amino acid sequences analysed in *Figure 4*.

---

fusion between NSs and HIV-induced CPSF6 puncta. To investigate this, we performed live imaging to track CPSF6-mNeonGreen and NSs in cells expressing endogenous SRRM2 fused with a Halo tag, using CRISPR Paint (courtesy of Roy Parker) (*Lester et al., 2021*). We fixed and labelled samples at different time points post-infection, ranging from 6 to 30 h.p.i. (*Figure 7A*, *Figure 7—figure supplements 1 and 2*). At 6 h.p.i., 27% of HIV-induced CPSF6 puncta were still individual, compared to only 9% at 30 h.p.i. (*Figure 7B*). Concurrently, 61% of HIV-induced CPSF6 puncta were fused with NS at 6 h.p.i., rising to 75% at 30 h.p.i. (*Figure 7B*). This indicates a progressive increase in the number of HIV-induced CPSF6 puncta fusing with NS over time (*Figure 7C*). Overall, we detected individual CPSF6 puncta (green) and NSs (red) that quickly fused, confirming that this fusion occurs within the two independent puncta, CPSF6 and SRRM2, rather than during the formation of HIV-induced CPSF6 puncta (*Figure 7A–C*, *Figure 7—videos 1 and 2*).

Taken together, these results suggest that HIV-induced CPSF6 puncta first form independently of NS and later fuse with NS, causing an enlargement of NS as part of the hijacking process by HIV-1.

## Role of SON and SRRM2 in the fusion and stabilization of HIV-induced CPSF6 puncta within NSs

Macrophage-like cells, THP-1, were depleted for SON or SRRM2 using AUM*silence* ASO technology (*Gao et al., 2024*; *Marasca et al., 2022*; *Mazzeo et al., 2024*; *Zhang et al., 2023*). The level of depletion of SON and SRRM2 was evaluated by immunofluorescence and western blot using antibodies against SON and SRRM2 (*Figure 8A*). Both depleted cells were analysed also for the presence of NS, labelled by SC35 (*Figure 8—figure supplement 1*). However, recent findings suggest that the primary target of the SC35 mAb is SRRM2 (*Ilik et al., 2020*). To confirm this, cells depleted of SRRM2 were labelled with antibodies against both SRRM2 and SC35 (*Figure 8—figure supplement 1*). We have observed that the reduction of SRRM2 resulted in a slight decrease in the mean intensity of SC35, whereas the depletion of SON (*Figure 8—figure supplement 1*) did not have the same effect.

Subsequently, we infected THP-1 depleted cells for SRRM2 and SON or control cells with HIV-1 and we fixed them at 48 hr post-infection for immunofluorescence. We calculated the percentage of CPSF6 puncta in HIV-infected THP-1 control cells (approximately 78%) and in HIV-infected THP-1 cells depleted for SRRM2 (about 43%) and SON (around 66%). Results from our experimental conditions indicate that the partial depletion of SRRM2 affects the formation of HIV-induced CPSF6 puncta, while the depletion of SON slightly reduces their establishment (*Figure 8B*).

## The IDR of SRRM2 is a crucial element for the fusion of HIV-induced CPSF6 puncta to the NSs

CPSF6 contains several disordered regions (*Di Nunzio et al., 2023*) as well as NS factors (*Ilik et al., 2020*). Previous studies have established the role of IDR of SRRM2 in the biogenesis of NS (*Ilik et al., 2020*).

Here, we investigate the role of IDRs within NS factors, with a specific emphasis on SRRM2, in the fusion and stabilization of HIV-induced CPSF6 puncta. To address this inquiry, we utilized previously published HEK293 cell lines generated using the CRISPaint system (*Lester et al., 2021*), comprising two distinct lines: HaloTag SRRM2, SRRM2 full-length (FL) (1–2748 aa) fused with Halo tag (insertion

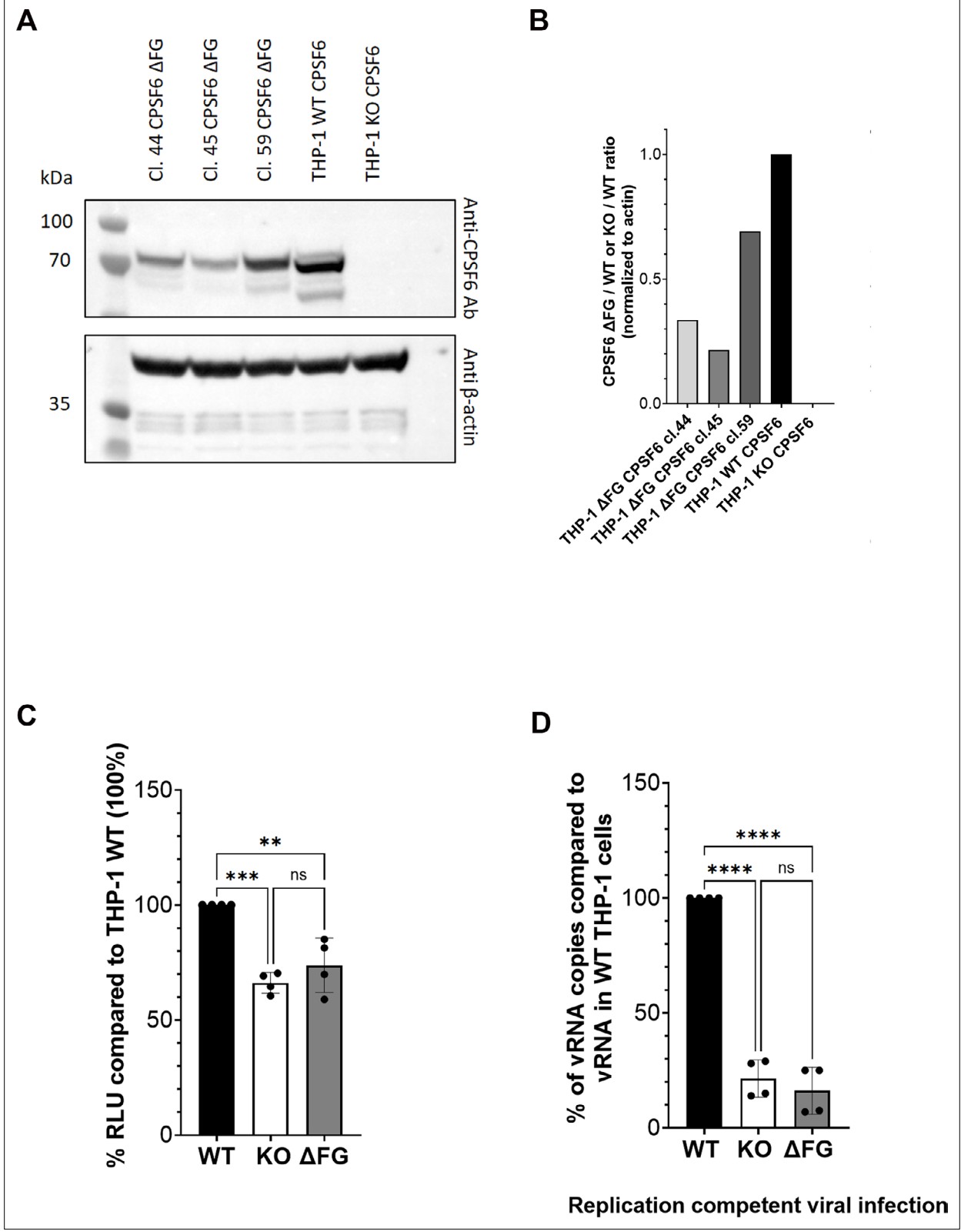

**Figure 5.** Role of FG motif in viral replication. (**A**) WB showing CPSF6 protein from several single clones derived from CPSF6 KO clone obtained upon complementation with CPSF6 ΔFG and normalized with beta-actin. (**B**) Quantification of the expression of CPSF6 ΔFG protein in different single clones compared to CPSF6 WT (value 1). (**C**) Infectivity assay using a single-round infectious virus carrying the cDNA of Luciferase as reporter gene. Values are expressed as % of RLU compared to WT cells. (**D**) Infectivity assay of a replication competent virus: vRNA from new viruses produced after infection of

*Figure 5 continued on next page*

*Figure 5 continued*

WT, CPSF6 KO, and CPSF6ΔFG cells was analysed and shown in the histograms as % of vRNA copies compared to vRNA in WT THP-1 cells considered 100%.

The online version of this article includes the following source data for figure 5:

**Source data 1.**

**Source data 2.**

**Source data 3.**

at amino acid [aa] 2708), and a cell line lacking the C-terminal IDR of SRRM2, known as ΔIDR HaloTag SRRM2 (1–429 aa, with halo insertion at aa 430).

As anticipated, the truncated form of SRRM2 displayed a more diffuse distribution within the nucleus, without recruitment to NS (*Figure 8—figure supplement 2*), and consequently lacked nuclear puncta (*Figure 8C*, top panels). On the other hand, the number of SON puncta was highly similar between the two cell lines (*Figure 8C*, panels on the bottom, *Figure 8—figure supplement 3*).

Subsequently, we quantified the formation of CPSF6 puncta in both cell lines infected with HIV-1. No significant difference in HIV-induced CPSF6 puncta formation was observed between HEK293 and HEK293 carrying the SRRM2 halo tag (~27%). However, a substantial reduction in HIV-induced CPSF6 puncta was evident in the cell line carrying the SRRM2 form that lacks the IDR (~11%) (*Figure 8D*). Collectively, these results underscore the pivotal role of the SRRM2 IDR in the stabilization of HIV-induced CPSF6 puncta through their fusion with NSs (*Figure 7A–C*).

## Discussion

HIV-1 capsid has transformed long-standing assumptions in the field. Previously considered an undruggable viral target due to the belief that it disassembled shortly after HIV entered target cells, whereas recent findings have revealed its essential role in nuclear import (*Ay and Di Nunzio, 2023*; *Blanco-Rodriguez and Di Nunzio, 2021*; *Blanco-Rodriguez et al., 2020*; *Chen et al., 2016*; *Taylor and Fassati, 2024*; *Yamashita and Emerman, 2004*; *Zila et al., 2021*). Notably, the first-in-class antiretroviral capsid inhibitor, Lenacapavir, has shown remarkable results in patients, demonstrating that the capsid, contrary to previous belief, can indeed be a viable therapeutic target. While Lenacapavir improves patients' quality of life with only two injections per year (*Link et al., 2020*; *Segal-Maurer et al., 2022*), no current antiretroviral drugs provide a cure. This may be due to our incomplete understanding of certain aspects of HIV biology. In this study, we shed light on the post-nuclear entry steps, a critical phase for the establishment of viral reservoirs, which represent the main barrier to a cure. Recent findings have shown that not only does the HIV viral capsid translocate through the NPC, but that the viral nuclear entry also enhances the formation of CPSF6 puncta. Additionally, it has been revealed that RT is completed within the nucleus (*Burdick et al., 2020*; *Dharan et al., 2020*; *Rensen et al., 2021*; *Scoca et al., 2023*; *Selyutina et al., 2020*). Furthermore, incoming viral RNA has been observed to be sequestered in nuclear niches in cells treated with the reversible RT inhibitor, NVP. When macrophage-like cells are infected in the presence of NVP, the incoming viral RNA is held within the nucleus (*Ay et al., 2025*; *Rensen et al., 2021*; *Scoca et al., 2023*). This scenario could also occur in patients receiving antiretroviral therapy. Notably, we found that pharmacological dismantling of CPSF6 puncta prevents the restoration of nuclear reverse transcription, indicating that HIV-induced CPSF6 puncta are critical for the viral life cycle. Conversely, THP-1 cells either depleted of CPSF6 or expressing a CPSF6 variant lacking the FG domain exhibit impaired viral replication. Understanding the biogenesis of these puncta could be a significant step towards deepening our knowledge of HIV biology and providing additional tools to combat this pandemic virus.

Here, we identify the disordered FG peptide essential for the binding with the viral capsid as the inducer of HIV-induced CPSF6 puncta. Notably, CPSF6 protein lacking the FG peptide is incapable of forming nuclear puncta. Moreover, we discovered that the two major IDRs of CPSF6, the MCDs and the LCRs, are dispensable for the formation of viral nuclear puncta. The MCDs of CPSF6 have been shown to provide cohesion for NS condensation (*Greig et al., 2020*). A study performed in HEK293T-derived CPSF6 knockout (CKO) cells (clone B8) complemented with a tagged CPSF6 reported that the MCD is critical for CPSF6 cluster formation (*Jang et al., 2024*). In contrast, we chose to work with a more

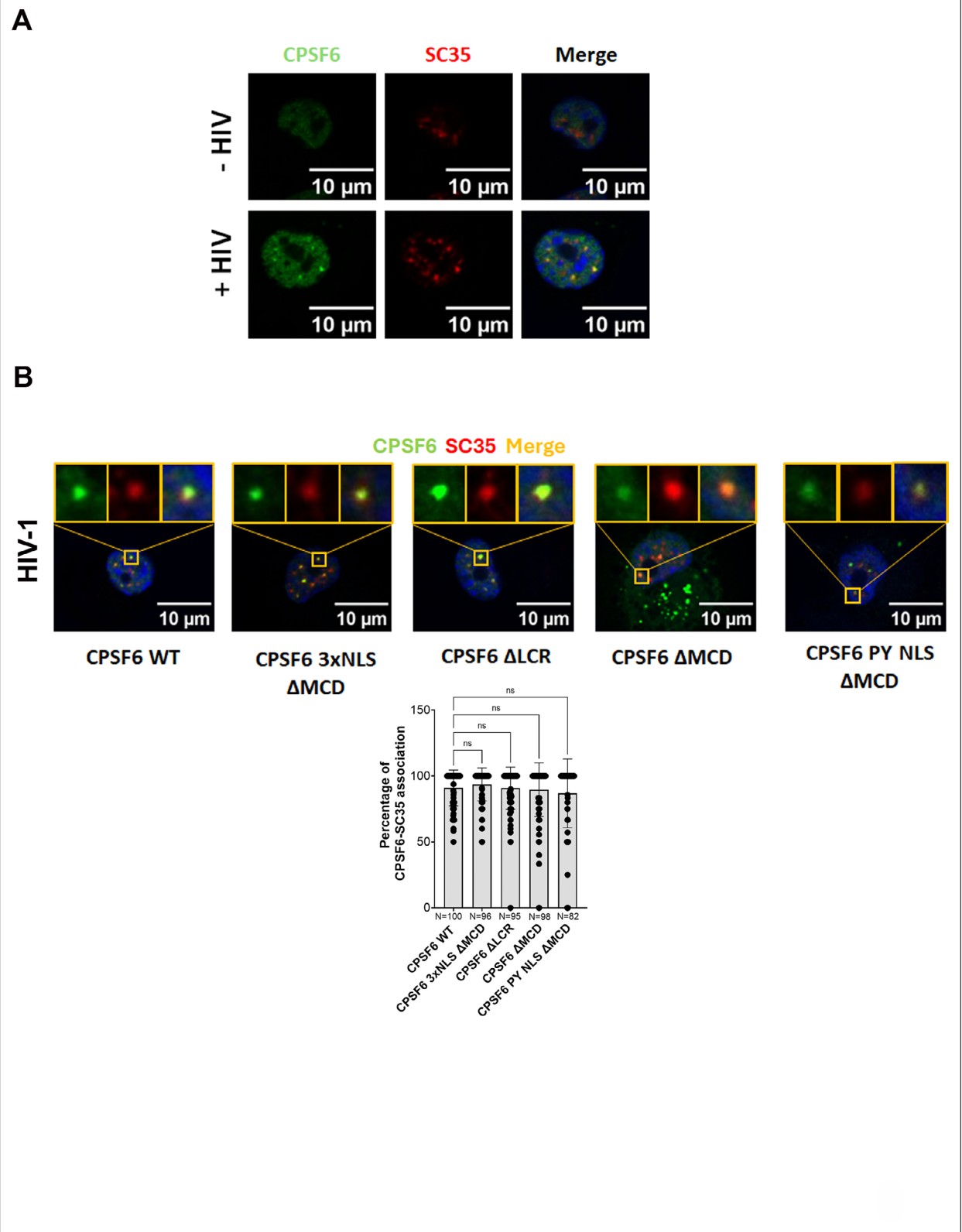

**Figure 6.** Depletion of mixed charge domain (MCD) or low-complexity region (LCR) does not affect the formation of HIV-induced CPSF6 puncta. (**A**) Epifluorescence microscopy images of both infected and non-infected differentiated THP-1 cells showing the presence of CPSF6 puncta only in the infected condition. CPSF6 and SC35 are labelled with anti-CPSF6 (green) and anti-SC35 (red) antibodies, respectively. Nuclei are stained with Hoechst (blue). Scale bar 10 µm. (**B**) Confocal microscopy images of THP-1 KO CPSF6 cells, differentiated for 3 days, transduced with CPSF6 lentiviral vector

*Figure 6 continued on next page*

*Figure 6 continued*

(MOI 1) (specifically WT CPSF6, CPSF6 ΔLCRs, CPSF6 ΔMCD with 3xNLS, without NLS, or with PY NLS) for 3 days and infected for 24 hr with VSV-G/HIV-1ΔEnvIN$_{HA}$ LAI (BRU) -vpx MOI 10 in presence of Nevirapine (10 μM). CPSF6 and nuclear speckles were labelled with anti-CPSF6 (green) and anti-SC35 (red) antibodies, respectively. Nuclei are stained with Hoechst (blue). Scale bar 10 μm. The percentage of CPSF6 puncta associated with SC35 per field of view is shown in the graph. *N* cells were counted in each condition and a one-way ANOVA statistical test with multiple comparison was performed; ns, p-value >0.05.

The online version of this article includes the following source data for figure 6:

**Source data 1.**

**Source data 2.**

**Source data 3.**

**Source data 4.**

**Source data 5.**

physiologically relevant cell line for HIV infection, macrophage-like cells, and we employed untagged CPSF6 proteins (WT and deletion mutants) to avoid potential artefacts introduced by protein tagging. Under our experimental conditions, the MCD did not appear to play a critical role in the formation of HIV-induced CPSF6 puncta. CPSF6 puncta induced by CPSF6 depleted for the MCD upon infection are highly similar to those formed by wild-type CPSF6, suggesting that MCDs do not play a significant role in this process. Additionally, the LCRs of CPSF6 also do not appear to influence puncta formation, as their depletion does not reduce the number of CPSF6 puncta. This indicates that neither the MCDs nor the LCRs are involved in CPSF6 puncta formation during HIV infection. Surprisingly, when we assessed the ability of CPSF6 domains to bind to the viral capsid, we observed that the deletion of LCRs increases CPSF6's ability to bind to the viral capsid. We hypothesize that a change in charges may alter the binding mechanism of CPSF6 when LCRs are absent. In scenarios where the FG motif is depleted, we observed a dramatic inhibition of HIV-induced CPSF6 puncta formation and a lack of binding to the viral core in vitro. On the flip side, the linkage between CPSF6 entities is facilitated by the FG peptides' interaction with certain hydrophobic CA pockets along adjoining hexamers (*Wei et al., 2022*). Therefore, it's conceivable that FG peptides, not involved in the capsid's binding, could coalesce similarly to FG-Nups. These undergo phase separation, forming condensates with NPC-like permeability barrier features (*Hülsmann et al., 2012*).

We have also identified HIV-induced CPSF6 puncta formation independent of NS at early stages post-infection, with a progressive increase in CPSF6 puncta colocalizing with NS over time. These results were obtained by focusing our studies on two scaffold proteins involved in NS biogenesis: SON and SRRM2. Over the past approximately 0.6–1.2 billion years of metazoan evolution, these two factors have undergone significant lengthening, unlike many other proteins involved in splicing. This extension primarily occurred within their IDRs, which are commonly associated with liquid–liquid phase separation and the formation of biomolecular condensates (*Rai et al., 2018*). Co-depleting SRRM2 with SON, or depleting SON in a cell line where it is deleted the intrinsically disordered C-terminus of SRRM2, abolished the formation of NSs (*Ilik et al., 2020*). However, the depletion of only one factor does not abolish NSs. Consistent with this finding, we observed that the depletion of SRRM2 does not affect the presence of SON nuclear puncta. Similar results were obtained with cells genetically modified to express the truncated form of SRRM2 lacking the IDRs. When the ΔIDR SRRM2-halo tag was detected by the halo ligand, there was no recruitment of the truncated SRRM2 form in NSs. However, if SC35 is used as a target for antibodies, it can still be detected, albeit with a much lower intensity signal than in cells expressing the full-length SRRM2. Additionally, we observed a significant reduction in the detection of HIV-induced CPSF6 puncta. The few CPSF6 puncta detected in these cells colocalized with the weak SC35 signal. These results suggest that the IDR of SRRM2 plays an important role in HIV-induced CPSF6 puncta stabilization, but rare puncta can appear, likely induced by redundant NS factors. This indicates that further investigation is needed to better understand the hijacking of NS by HIV.

Taken together, our results reveal the intricate interplay between individual CPSF6 domains and the viral capsid in dictating the nuclear fate of the virus. Concurrently, the IDR domain of SRRM2 contributes to the enlargement of particular NSs with the fusion of HIV-induced CPSF6 puncta. Lastly, these HIV-induced CPSF6 puncta necessitate FG peptide to engage the viral core.

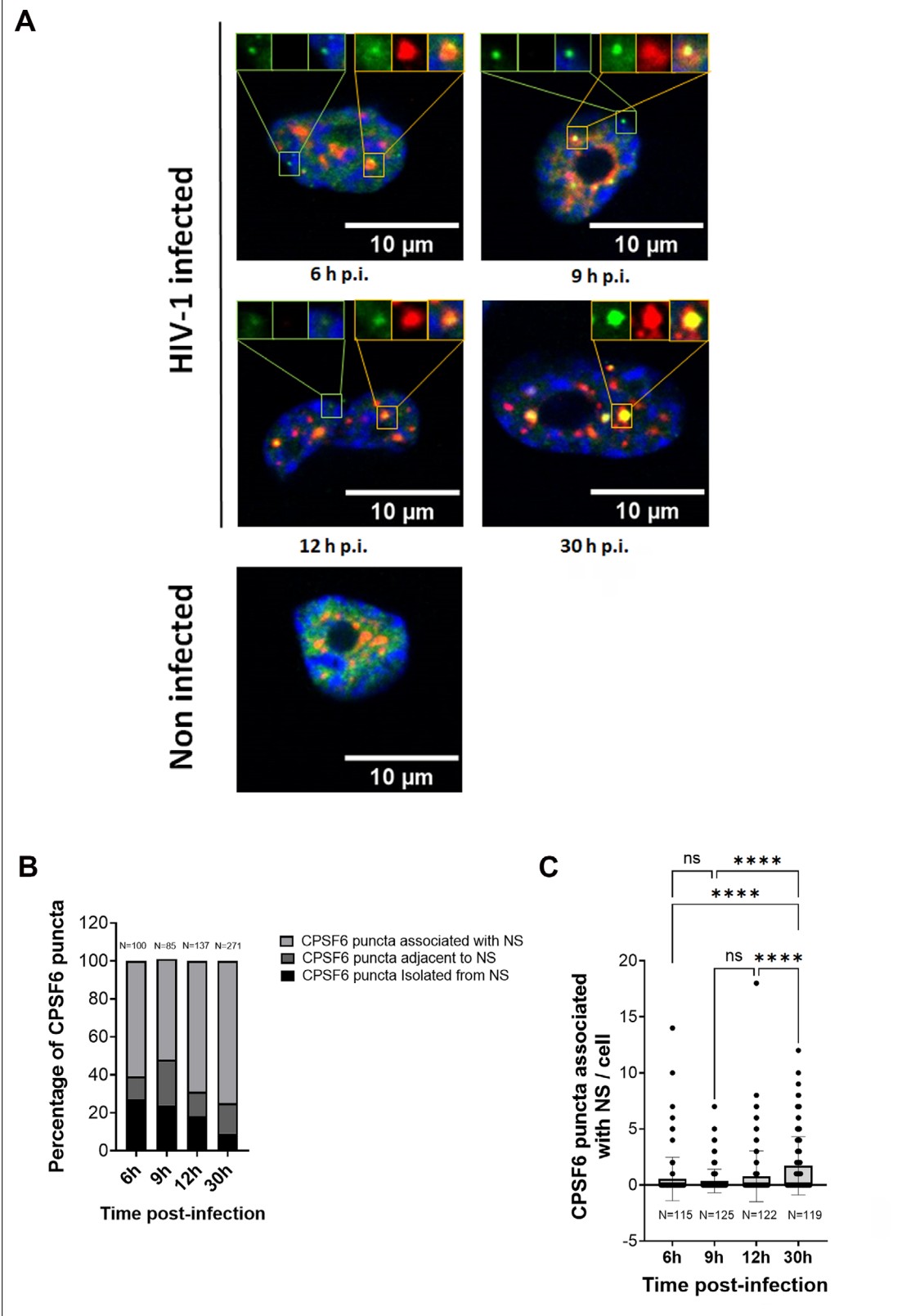

**Figure 7.** Dynamics of the HIV-induced CPSF6 puncta formation and their fusion with NSs. (**A**) Time course of infection of THP-1, 6, 9, 12, 30 h.p.i., or non-infected. Cells were stained with antibodies against CPSF6 (green) and SC35 (red). (**B**) The graph shows the percentage of CPSF6 puncta associated with nuclear speckle (NS) or adjacent to NS or isolated from NS at different time post-infection. (**C**) The graph shows the progression of CPSF6 puncta

*Figure 7 continued on next page*

*Figure 7 continued*

associated with NS during the time post-infection. *N* indicates the number of cells analysed. One-way ANOVA statistical test with multiple comparison was performed; ns, p-value >0.05; **** indicates a p-value <0.0001.

The online version of this article includes the following video, source data, and figure supplement(s) for figure 7:

**Source data 1.**

**Source data 2.**

**Source data 3.**

**Figure supplement 1.** THP-1 infected with VSV-G/HIV-1ΔEnvIN$_{HA}$ LAI (BRU) -vpx were labelled after 6, 9, 12, 30 h.p.i. with specific antibodies against CPSF6 (green) and SC35 (red), nuclei are stained with Hoechst (blue).

**Figure supplement 2.** Timelapse of *Figure 7—video 2* which recapitulates the key steps of the dynamics of HIV-induced-CPSF6 puncta biogenesis.

**Figure 7—video 1.** CPSF6 membraneless organelles (MLOs) can form independently from SRRM2 MLOs in HEK293 SRRM2 HaloTag cells.
https://elifesciences.org/articles/103725/figures#fig7video1

**Figure 7—video 2.** CPSF6 membraneless organelles (MLOs) can form independently from SRRM2 MLOs in HEK293 SRRM2 HaloTag cells.
https://elifesciences.org/articles/103725/figures#fig7video2

---

Overall, this study could provide insights into the understanding of viral invasion and persistence within the host.

## Materials and methods
### Cell lines

THP-1 cells (ATCC) are immortalized monocytic cells, which, once seeded, differentiate into macrophage-like cells under phorbol 12-myristate 13-acetate (PMA) treatment (160 nM). THP-1 cells were also engineered to knock out CPSF6. These cells were cultivated in RPMI 1640 medium supplemented with 10% foetal bovine serum (FBS) and 1% penicillin–streptomycin solution (100 U/ml). HEK293T cells (ATCC) are human embryonic kidney cells used to produce LVs. For *Figure 2*, we used two engineered HEK293 strains, Halo tagged SRRM2 HEK293 cells and Halo tagged SRRM2 ΔIDR HEK293 cells, kind gifts from Roy Parker's lab (*Lester et al., 2021*), characterized by the Halo tagged SRRM2 protein. In Halo tagged SRRM2 ΔIDR HEK293 cells, Parker's lab also deleted the SRRM2 sequence encoding for aa 430–2748. All the HEK293 strains were cultivated in Dulbecco's modified Eagle medium supplemented with 10% FBS and 1% penicillin–streptomycin (100 U/ml). All cells used in the study were tested for mycoplasma and they were negative.

### Bacteria strains

All *Escherichia coli* bacteria strains were grown in Luria-Bertani (LB) medium at 37°C. DHα competent cells and Stellar Competent Cells were used for molecular cloning, while *E. coli* One-Shot BL21star (DE3) cells were exploited for protein production.

### Plasmids

To express the WT CPSF6/WT CPSF6-mNeonGreen and the mutant CPSF6/mutant CPSF6-mNeonGreen clones, the correspondent coding sequences were engineered in pSICO plasmids. The two original plasmids used were pSICO CPSF6-mNeonGreen and pLPCX CPSF6 ADD2, a gift from Mamuka's lab. HIV-1ΔEnvIN$_{HA}$ LAI (BRU) plasmid encodes the ΔEnvHIV-1 LAI (BRU) viral genome where the IN protein is fused to the HA tag while pNL4.3 Δenv ΔNef IRES GFP plasmid encodes the ΔEnvHIV-1 NL4.3 viral genome and contains also a GFP sequence headed by an IRES. The pNL4.3 Δenv ΔNefLuc has the Luciferase cDNA as reporter gene.

### CRISPR–Cas9 knockout in THP-1 cells

To target CPSF6, three different crRNAs were used simultaneously (specific sequence: 5'-TCGGGCAAATGGCCAGTCAAAGG-3', 5'-AGGACGGGGCCGTTTTCCAGGGG-3', and 5'-CATGTAATCTCGGTCTTCTGGGG-3', all ordered from Integrated DNA Technologies, IDT). Pre-designed unspecific crRNA was used as control (IDT). crRNA and tracrRNA were resuspended in IDT Duplex Buffer according to the manufacturer's instructions. On the day of the nucleofection, duplexes were formed by mixing

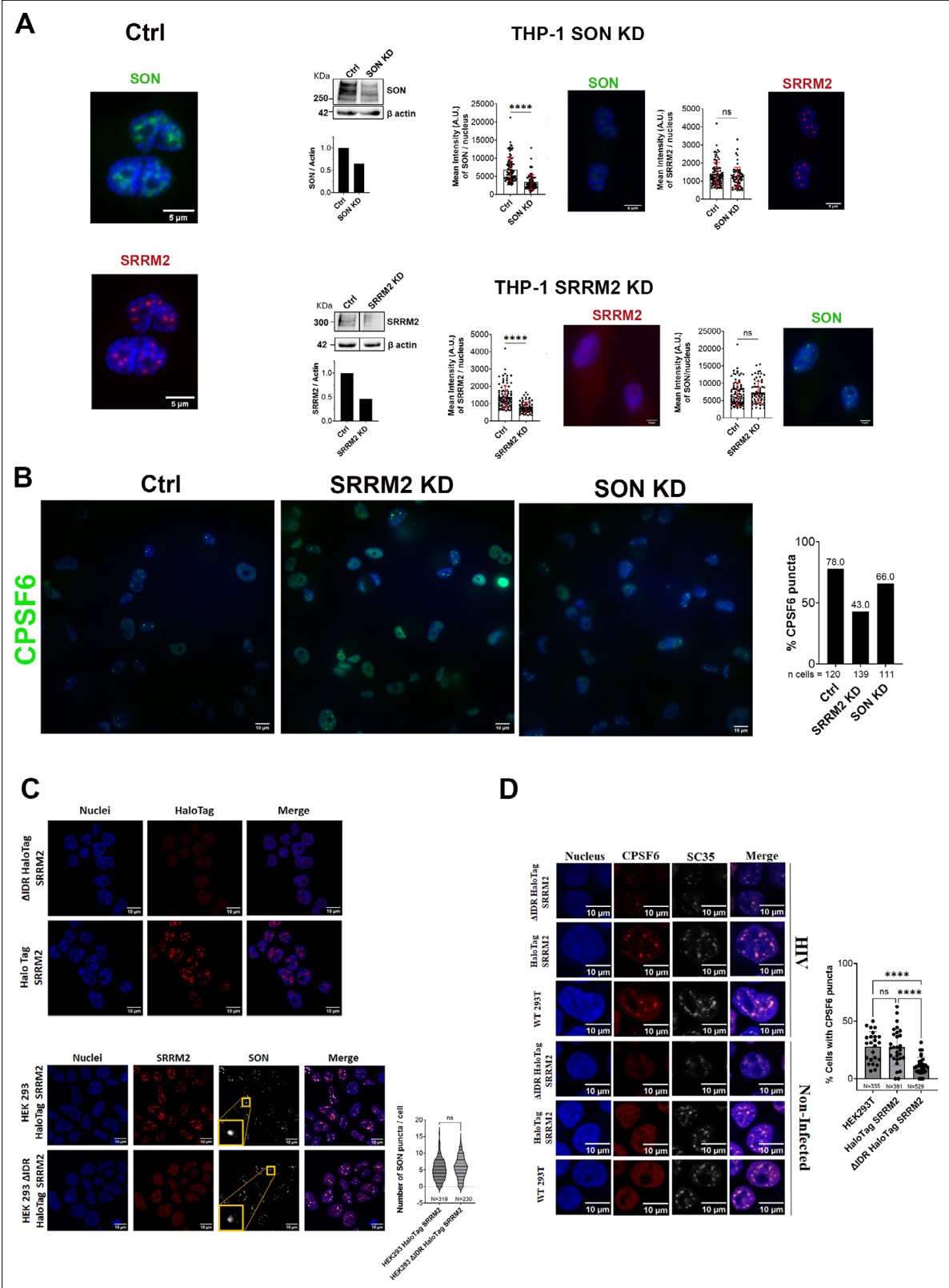

**Figure 8.** Role of SRRM2 and SON in the formation of HIV-induced CPSF6 puncta. (**A**) Depletion of SON and SRRM2 in THP-1 cells using AUM*silence* ASO technology. The degree of depletion is quantified by WB and the mean intensity through immunofluorescence using antibodies against SON and SRRM2, respectively. Scale bar 5 μm. (**B**) The percentage of CPSF6 puncta formation is quantified by IF in THP-1 cells knocked down for SON, SRRM2, and control (Ctrl) infected with HIV-1 (MOI 25) for 48 hr. CPSF6 is stained with an antibody against CPSF6 (green), and nuclei are stained with

*Figure 8 continued on next page*

*Figure 8 continued*

Hoechst (blue). The graph on the right reports the percentage of CPSF6 puncta calculated from more than 100 cells. Scale bar 10 µm. Experiments were performed at least twice. (**C**) (Top panels) Confocal microscopy images of ΔIDR HaloTag SRRM2 HEK293 and HaloTag SRRM2 HEK293 cells stained with the halo tag ligand (red), and nuclei (blue). Scale bar 10 µm. (Bottom panels) Confocal microscopy images of HaloTag SRRM2 HEK293 and ΔIDR HaloTag SRRM2 HEK293 cells, both labelled with anti-SRRM2 (red) and anti-SON (grey) antibodies. Nuclei are stained with Hoechst (blue). Scale bar 10 µm. Statistical studies are summarized in the violin plot which displays the distribution of the number of SON puncta per cell in the two conditions. *N* cells were counted and Kolmogorov–Smirnov test was performed, ns, p > 0.05. (**D**) Confocal microscopy images of HaloTag SRRM2 HEK293 and ΔIDR HaloTag SRRM2 HEK293 cells, either non-infected or infected for 24 hr with VSV-G/HIV-1ΔEnvIN$_{HA}$ LAI (BRU) (MOI 10) in the presence of Nevirapine (10 µM). CPSF6 and SC35 are labelled with anti-CPSF6 (red) and anti-SC35 (grey) antibodies, respectively. Nuclei are stained with Hoechst (blue). The plot shows the mean ± SD of the percentage of cells with CPSF6 puncta calculated in n fields of view (*n* = 24, 29, 32); *N* is the number of cells analysed for each of the three different cell lines; an unpaired *t*-test was performed, ****p < 0.0001; ns, p > 0.05. Scale bar 10 µm. Experiments were performed at least twice.

The online version of this article includes the following source data and figure supplement(s) for figure 8:

**Source data 1.**

**Source data 2.**

**Source data 3.**

**Source data 4.**

**Source data 5.**

**Source data 6.**

**Source data 7.**

**Source data 8.**

**Figure supplement 1.** Analysis of the mean intensity: SRRM2 depletion is confirmed using antibodies against both (**A**) SC35 and (**B**) SRRM2 by IF.

**Figure supplement 2.** HEK 293 HaloTag SRRM2 or ΔIDR were labelled with Halo ligand (red) and an anti-SON antibody (grey), nuclei are stained by Hoechst (blue).

**Figure supplement 3.** Multiple examples (**A–E**) of confocal microscopy images of HaloTag SRRM2 HEK293 and ΔIDR HaloTag SRRM2 HEK293 cells, either non-infected or infected for 24 hr with VSV-G/HIV-1ΔEnvIN$_{HA}$ LAI (BRU) (MOI 10) in the presence of Nevirapine (10 µM).

equimolar concentration of crRNA and tracrRNA, followed by 5 min annealing at 95°C. RNA duplexes were then mixed (1:2) with TrueCut Cas9 Protein v2 for 10 min at room temperature to generate ribonucleoprotein (RNP) complexes. $2 \times 10^5$ THP-1 cells were resuspended in P3 Primary Cell Nucleofector Solution, mixed with RNP and Alt-R Cas9 Electroporation Enhancer (90 pmol, IDT), and nucleofected in a 4D-Nucleofector System using the P3 Primary Cell 4D-NucleofectorX Kit S (program FI-110). After nucleofection, cells were seeded in complete RPMI medium with 20% FBS. Three days after nucleofection, cells were plated for clonal selection.

## Clonal selection of KO cell lines

Seventy-two hours post nucleofection, cells were diluted in RPMI medium containing 20% FBS and plated in five 96-well plates at 1 and 5 cells/well condition. After 1 month, selected microcolonies (50–100) are placed into 24-well plates. Once the wells were near confluence, cells were transferred into the well of a 6-well plate. After growing for another 1 month, cells were proceeded for western blot.

## Western blot

Proteins were extracted on ice from THP-1 cells using RIPA buffer (20 mM HEPES, pH 7.6, 150 mM NaCl, 1% sodium deoxycholate, 1% Nonidet P-40, 0.1% SDS, 2 mM EDTA, complete protease inhibitor), and protein concentration was quantified using a detergent-compatible (DC) protein assay (Pierce BCA Protein Assay Kit) with bovine serum albumin (BSA) as a standard. 90 µg of total protein lysate were loaded onto SDS–PAGE 4–12% Bis-Tris gels (Invitrogen); an Ab rabbit anti-CPSF6 (1:500) and an anti-rabbit HRP-conjugated (1:5000) were used for the detection of CPSF6, whereas the normalization was done by an Ab anti-actin HRP-conjugated (1:3000). Visualization was carried out using an ECL solution.

## AUMsilence ASO

AUMsilence™ 352 ASOs were synthesized by AUM BioTech, LLC (Philadelphia, USA). THP-1 negative control, THP-1 KD SRRM2, and THP-1 KD SON cells were differentiated with PMA (160 nM) for 48 hr then incubated 72 hr with 10 μM of a AUMsilence™ ASOs complementary to the mRNA of SRRM2 and SON, respectively (scramble control AUMscramble™). All cells were kept in an incubator at 37°C and 5% $CO_2$. These cells were cultivated in RPMI 1640 medium supplemented with 10% FBS and 1% penicillin–streptomycin solution (100 U/ml).

## Cloning

pSICO CPSF6-mNeonGreen was used to generate deletion mutants by excising different regions from the original sequence according to *Supplementary file 1*. The mutants were produced with and without the mNeonGreen tag, except for pSICO CPSF6-ΔMCD and pSICO CPSF6-ΔMCD PY NLS. All primers are specified in *Supplementary file 2*.

For pSICO CPSF6-mNeonGreen ΔLCR ADD2, CPSF6 ΔLCR ADD2 sequence was amplified by PCR from pLPCX CPSF6 ADD2 and specific primers were designed to add also BamHI restriction sites at the extremities. Phusion Flash High-Fidelity PCR Master Mix was used and the reaction was performed according to the manufacturer datasheet (100 ng template DNA and 56° annealing temperature). PCR products were treated with DpnI for 1 hr at 37°C and then digested with BamHI restriction enzyme at 37°C for 1 hr. 2 μl of the backbone pSICO CPSF6-mNeonGreen were digested with BamHI at 37°C for 3 hr and further treated with CIP. After gel extraction of the DNA and purification, insert and backbone were ligated with T4 Ligase for 2 hr at 22°C. 5 μl of the product were used to transform 50 μl of DH5α bacteria (30 min at 4°C, 45 s at 42°C, 2 min at 4°C, incubation in SOC medium for 1 hr at 37°C and plating on LB agar dishes).

All the other mutants were obtained using the In-FusionSnap Assembly protocol, which allowed us to amplify the original plasmid, deleting the small region of interest. The different primers used are reported in *Supplementary file 2*. As reported in the In-FusionSnap Assembly protocol , 5 ng of the original plasmid were amplified with PrimeSTARMax DNA polymerase in 35 PCR cycles (10 s at 98°C, 15 s at 55°C, 5 s/kb at 72°C). The PCR products were then digested for 1 hr at 37°C using DpnI to get rid of the original plasmid. After plasmid cleaning up, it was circularized through a ligation of 15 min at 50°C. 2.5 μl of products were used in Stellar Competent Cells' transformation, following the same procedure used for DH5a transformation, previously explained.

## Lentiviral vectors and viral productions

LVs and HIV-1 viruses were produced by transient transfection of HEK293T cells through calcium chloride co-precipitation. Co-transfection was performed as follows: for LVs, 10 μg of transfer vector, 10 μg of packaging plasmid (gag-pol-tat-rev), and 2.5 μg of pHCMV-VSV-G envelope plasmid; for VSV-G/HIV-1ΔEnvIN$_{HA}$ LAI (BRU) -vpx viruses and VSV-G/pNL4.3 Δenv ΔNef IRES GFP-vpx, 10 μg HIV-1ΔEnvIN$_{HA}$ LAI (BRU) plasmid or pNL4.3 Δenv ΔNef IRES GFP or pNL4.3 ΔenvΔRΔNef Luc, 2.5 μg of pHCMV-VSV-G plasmid and 3 μg of SIVMAC vpx (*Durand et al., 2013*) or full-length virus 10 μg HIV-1 NL4.3AD8 and 3 μg of SIVMAC vpx. After the collection of the supernatant 48 hr post-transfection, lentiviral particles were concentrated by ultracentrifugation for 1 hr at 22,000 rpm at 4°C and stored at −80°C. LVs and viruses were tittered by qPCR in HEK293T cells 3 days post-transduction.

## Cell transduction and infection

THP-1 ctrl CRISPR clone 2 cells and THP-1 (duplex1-2-3 CRISPR) KO clone 4 cells were differentiated with PMA (160 nM) for 72 hr then transduced with different mutants of CPSF6 for 72 hr (MOI = 1) and then infected for 30 hr with VSV-G/HIV-1ΔEnvIN$_{HA}$ LAI (BRU) -vpx (MOI = 10) in presence of Nevirapine (10 μM). The medium was always supplemented with PMA (160 nM).

For Halo tagged SRRM2 HEK 293 cells and Halo tagged SRRM2 ΔIDR HEK 293 cells, $2 \times 10^5$ cells were seeded on coverslips coated with polylysine in complete growth medium (DMEM, GlutaMAX-I, 10% FBS, and 1% P/S) and incubated at 37°C (5% $CO_2$) for 24 hr. Cells were then infected with the VSV-G/HIV-1ΔEnvIN$_{HA}$ LAI (BRU) (MOI 10) in complete growth medium supplemented with Nevirapine (10 mM) for 24 hr.

THP-1 control (scramble), THP-1 KD SRRM2 cells, and THP-1 KD SON cells were seeded on coverslips and differentiated with PMA (160 nM) for 72 hr. Then incubated for 48 hr with 10 μM of a FANA

ASOs (scramble control FANA (SCR-FANA), SRRM2-FANA, and SON-FANA). ASOs used in this study were designed and synthesized by AUM LifeTech (Philadelphia, PA, USA). Next, cells were infected with the pNL4.3 Δenv ΔNef IRES GFP-VPX (MOI 25) in complete growth medium and incubated at 37°C in 5% $CO_2$ for 4 days.

In the four time points experiment, THP-1 cells were differentiated with PMA (160 nM) for 72 hr. Cells were then infected with VSV-G/HIV-1ΔEnvIN$_{HA}$ LAI (BRU) -vpx (MOI = 10) in the presence of Nevirapine (10 µM), in complete growth medium supplemented with PMA (160 nM), and incubated at 37°C in 5% $CO_2$ for 6, 9, 12, 30 hr post-infection.

## Luciferase assay

The luciferase assay was performed using Luciferase Assay System (Promega, E4030). Infected differentiated THP-1 cells were washed with PBS and then incubated with Reporter Lysis Buffer at –80 °C overnight. After the thawing, cells were scraped, collected and centrifuged at 12,000 × g for 5 min to pull down all the cell debris. The supernatant was collected, and a part was used to react with the luciferase substrate. Data were acquired immediately at the luminometer.

## THP-1 infection with HIV-1 replication competent virus and viral RNA real-time quantification

THP-1 cells differentiate into macrophage-like cells within 48 hr under PMA treatment (160 nM). THP-1 cells were cultured at 37°C in 5% $CO_2$ in RPMI Medium 1640 supplemented with GlutaMAXTM-I, 10% FBS, 1% P/S (100 U/ml), MEM non-essential amino acid solution (0.1 mM), sodium pyruvate (1 mM), and HEPES buffer solution (10 mM). THP-1 cells differentiated into macrophages are infected with HIV-1 NL4.3AD8 with Vpx (MOI 6). Twenty-four hours after infection, the cells were washed and RPMI was added to the cells. Three days post-infection, the supernatant is collected and centrifuged. Total viral RNA was extracted from the supernatant via the QIAamp Viral RNA Mini Kit (QIAGEN, Hilden Germany) following the manufacturer's protocol. Total viral RNA was eluted into 60 µl of DEPC. vRNA is reverse transcribed into DNA with the Maxima H minus reverse transcriptase (Thermo Scientific, EP0752) following the manufacturer's protocol and the following primers were used (forward: GCCT CAATAAAGCTTGCCTTGA, reverse: TGACTAAAAGGGTCTGAGGGATCT). DNA copies for each sample were determined by qPCR and the SYBR green dye. The absolute DNA copies were calculated from a standard curve run in parallel using HIV-1 NL4.3AD8 plasmid. The qPCR cycle was as follows: 15 min at 95°C, 40 cycles (15 s at 95°C, 30 s at 58°C, 30 s at 72°C), 15 s at 95°C, 15 s at 55°C, 20 min melting curve until reaching 95°C, 15 s at 90°C.

## Immunofluorescence microscopy

### Immunostaining

On the day of fixation, the cells were washed with PBS and fixed with 4% PFA for 15 min. Cells were treated with glycine 0.15% for 10 min, permeabilized with 0.5% Triton X-100 for 30 min, and blocked with 1% BSA for 30 min. All antibody incubations were carried out at room temperature in a dark humid chamber, for 1 hr with primary antibodies and for 45 min with secondary antibodies. Washes between antibody incubations and antibody dilutions were done in 1% BSA.

Primary antibodies were diluted as follows: anti-HA 1:500, anti-CPSF6 1:400, anti-SC35 1:200, anti-SON 1:200, anti-SRRM2 1:200, secondary antibodies used were goat anti-rabbit Alexa-488 1:300 and donkey anti-rabbit Cy3 1:1000 for CPSF6, SON, and SRRM2, goat anti-rat Alexa-647 1:100 for IN-HA, goat anti-mouse Alexa-647 1:300 or goat anti-mouse Alexa-488 1:300 for SC35, donkey anti-mouse Alexa-647 1:300 for SRRM2.

Finally, cells were stained with Hoechst 33342 1:10,000 for 5 min. Coverslips were mounted on glass slides (Star Frost) with Prolong Diamond Antifade Mountant.

### HaloTag labelling

To detect the HaloTag in Halo tagged SRRM2 HEK 293 cells and Halo tagged SRRM2 ΔIDR HEK 293 cells, we used HaloTagTMR Ligand following the Technical Manual available at https://www.promega.com/protocols. Specifically, cells were incubated in DMEM supplemented with HaloTagTMR Ligand (5 µM) for 15 min at 37°C. The ligand-containing medium was then removed and replaced with an equal volume of 1X PBS, repeating the step twice and ending with warm complete medium. Cells

were incubated in complete culture medium for 30 min at 37°C. The medium was then removed and replaced with an equal volume of fresh warm culture medium.

### Immuno-RNA FISH

On the day of fixation, fixed cells were incubated in Permeabilization/Blocking buffer (1% BSA, 0.3% Triton X-100, 2 mM Vanadyl Ribonucleoside complexes (VRCs) in RNase-free PBS) for 1 hr before the antibodies' incubations. Antibodies were diluted in Permeabilization/Blocking buffer. After the primary and secondary antibody staining (respectively of 1 hr and 45 min) with the respective washes, cells were fixed for a second time in PFA 4% (in RNase-free PBS) for 10 min at room temperature with subsequent washes with RNase-free PBS. In the meantime, 40 pmol of primary smiFISH probes (*Tsanov et al., 2016*) 24 smiFISH probes designed against HIV-1 pol sequence (*Rensen et al., 2021*) were hybridized with 50 pmol of secondary FLAP probe conjugated to a Cy5 fluorophore (Cy5/AATG CATGTCGACGAGGTCCGAGTGTAA/Cy5Sp/) in 1X NEBuffer 3 (diluted in RNase-free $H_2O$) using a thermocycler. The programme setting follows: 3 min at 85°C, 3 min at 65°C, and 5 min at 25°C. FISH-probe solution was then diluted 1:50 in Hybridization buffer (90% Stellaris RNA-FISH Hybridization Buffer, 10% Deionized Formamide). After the samples were washed in Wash A buffer (20% Stellaris RNA-FISH Wash Buffer A, 10% Deionized Formamide, in RNase-free $H_2O$) at room temperature for 5 min, they were placed on parafilm, covered with 50 µl of FISH-probe solution in Hybridization buffer and incubated overnight at 37°C. The next day, cells were washed with Wash A buffer in the dark at 37°C for 30 min. Afterwards, the samples were incubated for 10 min in Hoechst 333342 diluted 1:10,000 in RNase-free $H_2O$ and then washed with Wash B buffer for 5 min at room temperature in the dark. Finally, the cells were washed in RNase-free $H_2O$ before the coverslips were mounted on microscopy slides using ProLongTM Diamond Antifade mounting medium. The mounting medium was cured overnight at room temperature under the chemical hood and away from light.

### Image acquisition

Images were acquired using a Zeiss LSM700 confocal inverted microscope, with a 63× objective (Plan Apochromat, oil immersion, NA = 1.4), using diode lasers at 405, 488, 555, and 639 nm for the respective fluorophores. A pixel size of 0.07 µm was used.

## Live imaging on HIV-1 infected Halo tagged SRRM2 HEK 293 cells

Halo tagged SRRM2 HEK293 cells were seeded and transduced with CPSF6-mNeonGreen lentiviral vector (MOI 0.5) for 24 hr. $0.3 \times 10^6$ cells were then transferred on poly-L-lysin coated Ibidi-dishes and infected with VSV-G/HIV-1ΔEnvIN$_{HA}$ LAI (BRU) (MOI = 40) in presence of Nevirapine (10 µM) for 2 hr. After having changed the medium and labelled with HaloTagTMR Ligand (5 µM) (see 'HaloTag labelling' section) and Hoechst 333342 diluted 1:80000 in complete DMEM medium, 4D movies were acquired.

3D movies were acquired using a Nikon Ti2-E Confocal Inverted Spinning Disk microscope, using a 63x objective (Plan Apochromat, oil immersion, NA = 1.4) and a sCMOS Hamamatsu camera, Orca Flash 4. Pixel size 6.5 µm, 2048 × 2044 pixel, QE 82%. For the Z stuck, a Z piezo stage was used, with 0.33 µm interval. For live imaging acquisitions, cells were placed in an environmental chamber with 37°C temperature, 5% $CO_2$ and 21% $O_2$.

## Protein expression and purification

pET-11a vectors were used to express the HIV-1 capsid protein. Point mutations, A14C and E45C, were introduced using the QuikChange II site-directed mutagenesis kit (Stratagene) according to the manufacturer's instructions. All proteins were expressed in *E. coli* one-shoot BL21star (DE3) cells (Invitrogen). Briefly, LB medium was inoculated with overnight cultures, which were grown at 30°C until mid log-phase ($A_{600}$, 0.6–0.8). Protein expression was induced with 1 mM isopropyl-β-d-thiogalactopyranoside overnight at 18°C. Cells were harvested by centrifugation at 5000 × $g$ for 10 min at 4°C, and pellets were stored at −80°C until purification. Purification of capsid was carried out as follows. Pellets from 2 l of bacteria were lysed by sonication (Qsonica microtip: 4420; A = 45; 2 min; 2 s on; 2 s off for 12 cycles), in 40 ml of lysis buffer (50 mM Tris pH = 8, 50 mM NaCl, 100 mM β-mercaptoethanol and Complete EDTA-free protease inhibitor tablets). Cell debris was removed by centrifugation at 40,000 × $g$ for 20 min at 4°C. Proteins from the supernatant were precipitated by

incubation with 1/3 of volume of saturated ammonium sulfate containing 100 mM β-mercaptoethanol for 20 min at 4°C and centrifugation at 8000 × g for 20 min at 4°C. Precipitated proteins were resuspended in 30 ml of buffer A (25 mM MES pH6.5, 100 mM β-mercaptoethanol) and sonicated two to three times (Qsonica microtip: 4420; A = 45; 2 min; 1 s on; 2 s off). The sample was dialysed three times in buffer A (2 hr, overnight, 2 hr). The sample was sonicated and diluted in 500 ml of buffer A and was chromatographed sequentially on a 5-ml HiTrap Q HP column and on a 5-ml HiTrap SP FF column (GE Healthcare), both pre-equilibrated with buffer A. The capsid protein was eluted from HiTrap SP FF column using a linear gradient from 0 to 2 M of NaCl. Absorbance at 280 nm was checked to take the eluted fraction that had higher protein levels. Pooled fractions were dialysed 3 times (2 h, overnight, 2 h) in storage buffer (25 mM MES, 2 M NaCl, 20 mM β-mercaptoethanol). Sample was concentrated using centricons to a concentration of 20 mg/ml and stored at –80 °C.

## Assembly of stabilized HIV-1 capsid tubes

1 ml of monomeric capsid (3 or 1 mg/ml) was dialysed in SnakeSkin dialysis tubing 10,000 MWCO (Thermo Scientific) against a buffer that is high in salt and contains a reducing agent (buffer 1: 50 mM Tris, pH 8, 1 M NaCl, 100 mM β-mercaptoethanol) at 4°C for 8 hr. Subsequently, the protein was dialysed against the same buffer without the reducing agent β-mercaptoethanol (buffer 2: 50 mM Tris, pH 8, 1 M NaCl) at 4°C for 8 hr. The absence of β-mercaptoethanol in the second dialysis allows formation of disulphide bonds between cysteine 14 and 43 inter-capsid monomers in the hexamer. Finally, the protein is dialysed against buffer 3 (20 mM Tris, pH 8.0, 40 mM NaCl) at 4°C for 8 hr. Assembled complexes were kept at 4°C up to 1 month.

## Capsid binding assay protocol

Human HEK293T cells were transfected for 24 hr with a plasmid expressing the specified CPSF6 variant tagged with mNeonGreen. Cell media was completely removed and cells were lysed in 300 μl of capsid binding buffer (CBB: 10 mM Tris, pH 8.0, 1,5 mM MgCl$_2$, 10 mM KCl) by scrapping off the plate. Cells were rotated at 4°C for 15 min and then centrifuged to remove cellular debris (21,000 × g, 15 min, 4°C). Cell lysates were incubated with stabilized HIV-1 capsid tubes for 1 hr at 25°C. Subsequently, stabilized HIV-1 capsid tubes were washed by pelleting the complexes by centrifugation at 21,000 × g for 2 min. Pellets were washed using resuspension in CBB or PBS. Pellets were resuspended in Laemmli buffer 1X and analysed by western blotting using anti-p24 or anti-mNeonGreen antibodies.

## Bioinformatics analysis of CPSF6

Intrinsic disorder propensity of CPSF6 was evaluated using the Rapid Intrinsic Disorder Analysis Online platform (RIDAO) (https://ridao.app/) designed to predict disordered residues and regions in a query protein based on its amino acid sequence (*Dayhoff and Uversky, 2022*). RIDAO yields results by combining the outputs of several commonly used per-residue disorder predictors, such as PONDR VLXT (*Romero et al., 2001*), PONDR VL3 (*Peng et al., 2006*), PONDR VLS2B (*Peng et al., 2005*), PONDR FIT (*Xue et al., 2010*), as well as IUPred2 (Short) and IUPred2 (Long)(*Dosztányi et al., 2005a*; *Dosztányi et al., 2005b*). RIDAO also computes a mean disorder score for each residue based on these. In the resulting intrinsic disorder profile, the disorder score of 0.5 is the threshold between order and disorder, where residues/regions above 0.5 are disordered, and residues/regions below 0.5 are ordered. The disorder score of 0.15 is the threshold between order and flexibility, where residues/regions with the disorder scores above 0.15 but below 0.5 are flexible, and residues/regions below 0.15 are highly ordered.

Amino acid compositions of the intrinsically disordered C-terminal domain (residues 261–358) of human CPSF6 and its different variants (CPSF6 ΔFG ΔLCR, CPSF6 ΔLCR, CPSF6 ΔFG, and CPSF6 ADD2 ΔLCR) were analysed to evaluate the relative abundance of prion-like low complexity region (LCR) defining uncharged, charged, and Pro residues in these protein regions. The corresponding values of the relative abundance of these residue groups were calculated by dividing numbers of prion-like LCR defining uncharged (Ala, Gly, Val, Phe, Tyr, Leu, Ile, Ser, Thr, Pro, Asn, Gln), charged (Asp, Glu, Lys, Arg), and Pro residues by total number of amino acids in the corresponding protein fragments. As references, we used the corresponding data for protein sequences deposited to the UniProtKB/Swiss-Prot database that provides information on the overall distribution of amino acids

in nature (*Bairoch et al., 2005*; PDB Select 25; *Berman et al., 2000*), which is a subset of structures from the Protein Data Bank with less than 25% sequence identity, biased towards the composition of proteins amenable to crystallization studies; and DisProt 3.4 that is comprised of a set of consensus sequences of experimentally determined disordered regions (*Sickmeier et al., 2007*). Per-residue intrinsic disorder propensities of the LCR-FG and ADD2-FG sequences were evaluated by PONDR VLXT (*Romero et al., 2001*), which is sensitive to local peculiarities of the amino acid sequences freely available at http://www.pondr.com/ (accessed on August 3, 2024). Linear distribution of the net charge per residue within the LCR-FG and ADD2-FG sequences was evaluated by CIDER (*Holehouse et al., 2017*), which is a webserver for the analysis of a wide range of the physicochemical properties encoded by IDP sequences freely available at http://pappulab.wustl.edu/CIDER (accessed on August 3, 2024). Secondary structure propensities of the LCR-FG and ADD2-FG sequences were evaluated by PSIPRED (*McGuffin et al., 2000*), which is a highly accurate secondary structure prediction method freely available to non-commercial users at http://globin.bio.warwick.ac.uk/psipred/ (accessed on August 3, 2024).

## Imaging and statistical analysis

All images were analysed using Fiji Software. More in detail, for the count of the NSs, a macro was computed to segment cellular nuclei and to select and count the NSs. For the nuclei segmentation, a Gaussian Blur with sigma = 2 was used and for the channel related to the NS, the set threshold was at 7000–46,012 (min and max). For the counting, a 'size = 10 – Infinity summarize' was used for the nuclei, whereas for the NSs the size was reduced to 'size = 0 – Infinity summarize'.

For live imaging analysis, Arivis software was used to reconstruct the 3D movies.

All data were statistically analysed with GraphPad Prism 9 (GraphPad Software, La Jolla, California USA, https://www.graphpad.com). Calculations were performed and figures were drawn using Excel 365 or GraphPad Prism 8.0. Statistical analysis was performed, with Wilcoxon matched paired *t*-tests or Mann–Whitney unpaired *t*-tests. Spearman correlation coefficients (*r*) were calculated using GraphPad Prism.

## Acknowledgements

FDN is supported by the Institut Pasteur and ANRS grants (ECTZ192036 and ECTZ137593), ANR-PRCI grant, Sidaction grant. CT is supported by fellowships Sidaction, ANR-PRCI. SA is supported by ANRS fellowship ECTZ204694. FD-G, BC, MR, CL, and CB are supported by NIH Grants R01AI087390 and R01AI150455. We gratefully acknowledge the UtechS Photonic BioImaging platform (Imagopole), C2RT at Institut Pasteur. We thank the NIH AIDS Reagents programme for supporting us with precious reagents and Addgene.

## Additional information

### Funding

| Funder | Grant reference number | Author |
| --- | --- | --- |
| Agence Nationale de Recherches sur le Sida et les Hépatites Virales | ECTZ192036 | Francesca Di Nunzio |
| Agence Nationale de Recherches sur le Sida et les Hépatites Virales | ECTZ137593 | Francesca Di Nunzio |
| Agence Nationale de la Recherche | | Francesca Di Nunzio |
| Agence Nationale de Recherches sur le Sida et les Hépatites Virales | ECTZ204694 | Selen Ay |

| Funder | Grant reference number | Author |
|---|---|---|
| Agence Nationale de Recherches sur le Sida et les Hépatites Virales | ECTZ290008 | Chiara Tomasini |
| National Institutes of Health | R01AI087390 | Felipe Diaz-Griffero |
| National Institutes of Health | R01AI150455 | Felipe Diaz-Griffero |

The funders had no role in study design, data collection, and interpretation, or the decision to submit the work for publication.

## Author contributions

Chiara Tomasini, Data curation, Formal analysis, Investigation, Methodology, Project administration; Celine Cuche, Data curation, Formal analysis, Methodology; Selen Ay, Maxence Collard, Bin Cui, Shaoni Bhattacharjee, Bruno Tello-Rubio, Julian Buchrieser, Charlotte Luchsinger, Cinzia Bertelli, Formal analysis; Mohammad Rashid, Vladimir Uversky, Data curation, Formal analysis; Felipe Diaz-Griffero, Data curation, Formal analysis, Supervision, Funding acquisition, Investigation, Methodology; Francesca Di Nunzio, Conceptualization, Resources, Data curation, Formal analysis, Supervision, Funding acquisition, Validation, Investigation, Visualization, Methodology, Writing – original draft, Project administration, Writing – review and editing

## Author ORCIDs

Felipe Diaz-Griffero (iD) https://orcid.org/0000-0002-7021-6152
Francesca Di Nunzio (iD) https://orcid.org/0000-0003-2879-3164

Reviewer #1 (Public review): https://doi.org/10.7554/eLife.103725.3.sa1
Reviewer #2 (Public review): https://doi.org/10.7554/eLife.103725.3.sa2
Reviewer #3 (Public review): https://doi.org/10.7554/eLife.103725.3.sa3
Author response https://doi.org/10.7554/eLife.103725.3.sa4

## Additional files

### Supplementary files

Supplementary file 1. Table (Table 1) describing the deleted regions of CPSF6 protein in different constructs expressing CPSF6 deletion mutants.

Supplementary file 2. Table (table 2) describing oligonucleotides used for cloning, qPCR, or ASOs.

MDAR checklist

### Data availability

From Figure 1 to Figure 8, the source data contain the numerical data used to generate the figures.

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

## Appendix 1

**Appendix 1—key resources table**

| Reagent type (species) or resource | Designation | Source or reference | Identifiers | Additional information |
|---|---|---|---|---|
| Strain, strain background (*Escherichia coli*) | DH5α | Thermo Fisher Scientific | #EC0112 | |
| Strain, strain background (*Escherichia coli*) | BL21(DE3) | Invitrogen | #C601003 | |
| Strain, strain background (*Escherichia coli*) | Stellar Competent Cells | Takara | #636763 | |
| Cell line (*Homo sapiens*) | HEK293T | ATCC | #CRL-3216 | |
| Cell line (*Homo sapiens*) | Halo tagged SRRM2 HEK293 cells | *Lester et al., 2021* | | Derived from ATCC Cat# CRL-3275, RRID:CVCL_DA04 |
| Cell line (*Homo sapiens*) | Halo tagged SRRM2 ΔIDR HEK293 cells | *Lester et al., 2021* | | Derived from ATCC Cat# CRL-3275, RRID:CVCL_DA04 |
| Cell line (*Homo sapiens*) | THP-1 | ATCC | #TIB-202 | |
| Cell line (*Homo sapiens*) | THP-1 KO CPSF6 | This paper | | |
| Antibody | Rat monoclonal anti-HA | Roche | #11867423001 | 1:500 |
| Antibody | Mouse monoclonal anti-p24 | NIH reagent | #NIH183-H12-5C | 1:400 |
| Antibody | Rabbit monoclonal anti-CPSF6 | Novus Biologicals | #NBP1-85676 | IF 1:400 WB 1:500 |
| Antibody | Mouse monoclonal anti-SC35 | Abcam | #ab11826 | 1:200 |
| Antibody | Rabbit monoclonal anti-SON (human) | Sigma-Aldrich | #HPA031755 | IF 1:200 WB 1:500 |
| Antibody | Rabbit monoclonal anti-SRRM2 | Sigma-Aldrich | #HPA066181 | IF 1:200 WB 1:500 |
| Antibody | Donkey polyclonal anti-mouse Cy3 | Jackson Lab | #715-165-150 | 1:100 |
| Antibody | Goat polyclonal anti-rat Alexa FluorTM 647 | Invitrogen | #A21247 | 1:100 |
| Antibody | Goat polyclonal anti-rat Alexa FluorTM 555 | Invitrogen | #A21434 | 1:300 |
| Antibody | Goat polyclonal anti-rabbit Alexa FluorTM 488 | Invitrogen | #A32731 | 1:300 |
| Antibody | Goat polyclonal anti-mouse Alexa FluorTM 647 | Invitrogen | #A21235 | 1:300 |
| Antibody | Mouse monoclonal anti-β actin HRP-conjugated | Abcam | #8226 | 1:3000 |
| Antibody | Mouse polyclonal anti-rabbit HRP-conjugated | Santa Cruz | sc2357 | 1:5000 |

*Appendix 1 Continued on next page*

*Appendix 1 Continued*

| Reagent type (species) or resource | Designation | Source or reference | Identifiers | Additional information |
|---|---|---|---|---|
| Recombinant DNA reagent | HIV-1 (BRU) ΔEnv IN(HA) (plasmid) | *Petit et al., 1999*; *Petit et al., 2000* | | Used to produce single-round virus |
| Recombinant DNA reagent | HIV-1 NL4.3 ΔEnv ΔVpr Luc | *Di Nunzio, 2013* | | Used to produce single-round virus |
| Recombinant DNA reagent | HIV-1 NL4.3 AD8 | NIH-AIDS Reagent Program | #11346 | Used to produce replicative virus |
| Recombinant DNA reagent | SIV$_{MAC}$ Vpx (plasmid) | *Durand et al., 2013* | | Used in lentiviral production |
| Recombinant DNA reagent | pCMV-VSV-G (plasmid) | Addgene | Plasmid #8454 | Used in lentiviral vectors' production |
| Recombinant DNA reagent | pSD-GP-NDK | PlasmidFactory | | Used in lentiviral vectors' production |
| Recombinant DNA reagent | pSICO CPSF6-mNeonGreen (plasmid) | Addgene | Plasmid #167587 | Used in lentiviral vectors' production |
| Recombinant DNA reagent | pLPCX CPSF6 ADD2 (plasmid) | *Wei et al., 2022* | | |
| Recombinant DNA reagent | pSICO-CPSF6 mutants (plasmids) | This paper | *Supplementary file 2* | Used in lentiviral vectors' production |
| Sequence-based reagent | Primary smiFISH probes against HIV-1 pol (24) | *Scoca et al., 2023* | | |
| Sequence-based reagent | Cy5 FLAP (RNA FISH secondary probe) | Eurofins Genomics | Probe | AATGCATG TCGACGAG GTCCGAGT GTAA |
| Sequence-based reagent | qPCR and cloning primers | This paper | *Supplementary file 1* | |
| Sequence-based reagent | AUMsilenceTM 352 ASOs against SRRM2 and SON | AUM BioTech | *Supplementary file 1* | |
| Sequence-based reagent | Alt-R CRISPR–Cas9 tracrRNA | IDT | # 1072532 | |
| Sequence-based reagent | Alt-R CRISPR–Cas9 crRNA | IDT | Materials and methods | |
| Peptide, recombinant protein | BamHI-HF | New England BioLabs | #R3136S | Used for the cloning |
| Peptide, recombinant protein | Quick CIP | New England BioLabs | #M0525S | Used for the cloning |
| Peptide, recombinant protein | Alt-R S.p. Cas9 V3 | IDT | #10007806 | |
| Peptide, recombinant protein | Transcriptase inverse Maxima H Minus (200 U/µl) | Thermo Fisher Scientific | #EP0752 | |
| Peptide, recombinant protein | T4 Ligase | Thermo Fisher Scientific | #EL0011 | |
| Commercial assay or kit | In-Fusion Snap Assembly Master Mix | Takara | #638947 | |
| Commercial assay or kit | Luciferase Assay System | Promega | #E4030 | |

*Appendix 1 Continued on next page*

*Appendix 1 Continued*

| Reagent type (species) or resource | Designation | Source or reference | Identifiers | Additional information |
|---|---|---|---|---|
| Commercial assay or kit | P3 Primary Cell 4D-Nucleofector X Kit S | Lonza | # V4XP-3032 | |
| Commercial assay or kit | NucleoSpin Plasmid | Macherey-Nagel | #740588.50 | |
| Commercial assay or kit | NucleoSpin Plasmid | Macherey-Nagel | #740416.10 | |
| Commercial assay or kit | PCR clean-up and gel extraction | Macherey-Nagel | #740609.50 | |
| Commercial assay or kit | QIAamp Viral RNA Mini Kit | QIAGEN | #52904 | |
| Commercial assay or kit | Pierce BCA Protein Assay Kits | Thermo Scientific | #23225 | |
| Chemical compound, drug | Bovine serum albumin (BSA) | Sigma-Aldrich | #A9647 | |
| Chemical compound, drug | Bradford | Bio-Rad | #500-0006 | |
| Chemical compound, drug | Calcium chloride solution | Sigma-Aldrich | #21115 | |
| Chemical compound, drug | cOmplete, EDTA free (tablet) | Sigma-Aldrich | #11873580001 | |
| Chemical compound, drug | Deionized Formamide | Bio Basic | #FB0211 | |
| Chemical compound, drug | ECL solution | Cytiva | #RPN2232 | |
| Chemical compound, drug | Ethanol Absolute | Fisher BioReagents | #BP2818-500 | |
| Chemical compound, drug | Ethylenediamine tetra-acetic acid disodium salt solution (EDTA) | Sigma-Aldrich | #E7889 | |
| Chemical compound, drug | Foetal bovine serum (FBS) | Serana | #S-FBS-SA-015 | |
| Chemical compound, drug | Glycine | Sigma | #G8898 | |
| Chemical compound, drug | HaloTagTMR Ligand | Promega | #G8252 | 1:1000 in cell culture medium for live imaging acquisitions |
| Chemical compound, drug | HEPES-buffered saline, pH 7.0 | Fisher Scientific | #J62623.AK | Used 1:2 for transfection |
| Chemical compound, drug | HEPES solution | Gibco | #15630 | Used for cell culture |
| Chemical compound, drug | Hoechst 33342 | Invitrogen | #H3570 | 1:10,000 in water for fixed cells 1:80,000 in cell culture medium for live imaging acquisitions |
| Chemical compound, drug | NEBuffer 3 | New England BioLabs | #B7003S | |

*Appendix 1 Continued on next page*

*Appendix 1 Continued*

| Reagent type (species) or resource | Designation | Source or reference | Identifiers | Additional information |
|---|---|---|---|---|
| Chemical compound, drug | Nevirapine | Sigma-Aldrich | #SML0097 | |
| Chemical compound, drug | NuPAGE Bis-Tris Mini Protein Gels, 4–12% | Invitrogen | #NP0322BOX | |
| Chemical compound, drug | Paraformaldehyde 32% | Electron Microscopy Sciences | #15714 | |
| Chemical compound, drug | Penicillin–Streptomycin (P/S) | Gibco | #15140 | |
| Chemical compound, drug | PF74 | Sigma | #SML0835 | |
| Chemical compound, drug | Phorbol 12-myristate 13-acetate (PMA) | Sigma | #P8139 | |
| Chemical compound, drug | Poly-L-lysine solution | Sigma | #P4707 | |
| Chemical compound, drug | Precision Plus Protein Western C, Standard solution | Bio-Rad | #1610376 | |
| Chemical compound, drug | ProLong Diamond Antifade Mountant | Thermo Fisher Scientific | #P36970 | |
| Chemical compound, drug | Reducing Agent 20X | Bio-Rad | #1610792 | |
| Chemical compound, drug | RIPA buffer | Sigma-Aldrich | #R0278 | |
| Chemical compound, drug | RNaseZAP | Sigma-Aldrich | #R2020 | |
| Chemical compound, drug | Running buffer 20X | Invitrogen | #NP0001 | |
| Chemical compound, drug | Sample Buffer 4X | Bio-Rad | #1610791 | |
| Chemical compound, drug | Stellaris RNA FISH Hybridization Buffer | Biosearch Technologies | #SMF-HB1 | |
| Chemical compound, drug | Stellaris RNA FISH Wash Buffer A | Biosearch Technologies | #SMF-WA1 | |
| Chemical compound, drug | Stellaris RNA FISH Wash Buffer B | Biosearch Technologies | #SMF-WB1 | |
| Chemical compound, drug | SuperScript III Platinum SYBR Green One-Step qRT-PCR | Invitrogen | #11736059 | |
| Chemical compound, drug | Transfer buffer 20X | Invitrogen | #NP0006-1 | |
| Chemical compound, drug | Triton X-100 | Sigma | #T8787 | |
| Chemical compound, drug | Trypsin 0.05%-EDTA (1X) | Gibco | #253000 | |
| Chemical compound, drug | TWEEN 20 | Sigma-Aldrich | #P1379 | |
| Chemical compound, drug | Phusion Flash High-Fidelity PCR Master Mix | Thermo Scientific | #F-548S | |

*Appendix 1 Continued on next page*

*Appendix 1 Continued*

| Reagent type (species) or resource | Designation | Source or reference | Identifiers | Additional information |
|---|---|---|---|---|
| Chemical compound, drug | UltraPure Distilled Water (H$_2$O) | Invitrogen | #10977 | |
| Chemical compound, drug | Vanadyl Ribonucleoside Complexes (VRC) | Sigma | #94742 | |
| Software, algorithm | ZEISS arivis Scientific Image Analysis | ZEISS | https://www.arivis.com/ | |
| Software, algorithm | CIDER | *Holehouse et al., 2017* | http://pappulab.wustl.edu/CIDER | |
| Software, algorithm | Fiji | *Schindelin et al., 2012* | https://imagej.net/software/fiji/ | |
| Software, algorithm | Icy | *de Chaumont et al., 2012* | https://icy.bioimageanalysis.org/download/ | |
| Software, algorithm | PONDR VLXT | *Romero et al., 2001* | | |
| Software, algorithm | Prism9 | GraphPad Software | https://www.graphpad.com/scientific-software/prism/ | |
| Software, algorithm | PSIPRED | *McGuffin et al., 2000* | http://globin.bio.warwick.ac.uk/psipred/ | |
| Software, algorithm | Rapid Intrinsic Disorder 696 Analysis Online platform (RIDAO) | *Dayhoff and Uversky, 2022* | https://ridao.app/ | |
| Other | BD SYRINGE 60 ml (no needle) | Dutscher | #309653 | |
| Other | Centrifuge | Thermo Fisher Scientific | Sorvall ST4 Plus | |
| Other | Confocal microscope | Zeiss | LSM700 inverted | |
| Other | Confocal microscope 63x objective | Zeiss | Objective Plan-Apochromat 63x/1.4 Oil DIC M27 | |
| Other | Corning black 96 Well Solid Polystyrene Microplate | Merck | #CLS3916 | |
| Other | Enspire 2300 | Perkin Elmer | | |
| Other | Falcon Cell culture 24-well plate | Dutscher | #353047 | |
| Other | Glasstic Slide 10 with Grids | KOVA | #87144E | |
| Other | HiTrapTM Q HP column | Cytiva | #17115401 | |
| Other | HiTrapTM SP FF column | Cytiva | #17515701 | |
| Other | Ibidi micro dishes 35 mm high | Ibidi | #81158 | |
| Other | Minisart NML Syringe Filter 0.45 µm | Sartorius | #16555 | |

*Appendix 1 Continued on next page*

*Appendix 1 Continued*

| Reagent type (species) or resource | Designation | Source or reference | Identifiers | Additional information |
|---|---|---|---|---|
| Other | Open-Top Thin wall Polypropylene Conical Tube, 31.5 ml, 25 × 89 mm | Beckman Coulter, Inc | #358126 | |
| Other | OPTILUX Petri dish – 100 × 20 mm | Dutscher | #353003 | |
| Other | Precision cover glasses 12 mm Æ thickness No. 1.5H | Marienfeld | #0117520 | |
| Other | Refrigerated benchtop centrifuge | Eppendorf | Centrifuge 5415 R | |
| Other | Rotor for LE-80K | Beckman Coulter, Inc | SW32Ti | |
| Other | Syngene GeneGenius Bio Imaging System Gel Documentation UV Transilluminator | Syngene | | |
| Other | Star-Frost slides 76 × 26 mm | Dutscher | #100204 | |
| Other | Thermocycler | Eppendorf | Mastercycler nexus | |
| Other | Thermocycler (qPCR) | Eppendorf | Mastercycler realplex2 | |
| Other | Ultracentrifuge | Beckman Coulter, Inc | Optima LE-80K | |

