## [Editor Report · eLife Assessment]

This is a **valuable** study that presents **convincing** evidence on the genesis of the CPSF6 condensates that form upon HIV-1 infection and the specific molecular determinants involved in their formation, as well as their interactions with SRRM. The study could be strengthened by assessing the relevance of their findings to infection, and in particular, with reverse transcription and gene expression

---

## [Referee Report · Reviewer #1 (Public review)]

In recent years, our understanding of the nuclear steps of the HIV-1 life cycle has made significant advances. It has emerged that HIV-1 completes reverse transcription in the nucleus and that the host factor CPSF6 forms condensates around the viral capsid. The precise function of these CPSF6 condensates is under investigation, but it is clear that the HIV-1 capsid protein is required for their formation. This study by Tomasini et al. investigates the genesis of the CPSF6 condensates induced by HIV-1 capsid, what other co-factors may be required and their relationship with nuclear speckels (NS). The authors show that disruption of the condensates by the drug PF74, added post-nuclear entry, blocks HIV-1 infection, which supports their functional role. They generated CPSF6 KO THP-1 cell lines, in which they expressed exogenous CPSF6 constructs to map by microscopy and pull down assays the regions critical for the formation of condensates. This approach revealed that the LCR region of CPSF6 is required for capsid binding but not for condensates whereas the FG region is essential for both. Using SON and SRRM2 as markers of NS, the authors show that CPSF6 condensates precede their merging with NS but that depletion of SRRM2, or SRRM2 lacking the IDR domain, delays the genesis of condensates, which are also smaller.

The study is interesting and well conducted and defines some characteristics of the CPSF6-HIV-1 condensates. Their results on the NS are valuable. The data presented are convincing.

I have two main concerns.

Firstly, the functional outcome of the various protein mutants and KOs is not evaluated. Although Figure 1 shows that disruption of the CPSF6 puncta by PF74 impairs HIV-1 infection, it is not clear if HIV-1 infection is at all affected by expression of the mutant CPSF6 forms (and SRRM2 mutants), or KO/KD of the various host factors. The cell lines are available, and so it should be possible to measure HIV-1 infection and reverse transcription. Secondly, the authors have not assessed if the effects observed on the NS impact HIV-1 gene expression, which would be interesting to know given that NS are sites of highly active gene transcription. With the reagents at hand, it should be possible to investigate this too.

Comments on revisions:

The revised version of this paper addresses my concerns.

---

## [Referee Report · Reviewer #2 (Public review)]

Summary:

HIV-1 infection induces CPSF6 aggregates in the nucleus that contain the viral protein CA. The study of the functions and composition of these nuclear aggregates have raised considerable interest in the field, and they have emerged as sites in which reverse transcription is completed and in the proximity of which viral DNA becomes integrated. In this work, the authors have mutated several regions of the CPSF6 protein to identify the domains important for nuclear aggregation, in addition to the already-known FG-region; they have characterized the kinetics of fusion between CPSF6 aggregates and SC35 nuclear speckles and have determined the role of two nuclear speckle components in this process (SRRM2, SUN2).

Strengths:

The work examines systematically the domains of CPSF6 of importance for nuclear aggregate formation in an elegant manner in which these mutants complement an otherwise CPSF6-KO cell line. In addition, this work evidences a novel role for the protein SRRM2 in HIV-induced aggregate formation, overall advancing our comprehension of the components required for their formation and regulation.

---

## [Referee Report · Reviewer #3 (Public review)]

In this study, the authors investigate the requirements for the formation of CPSF6 puncta induced by HIV-1 under a high multiplicity of infection conditions. Not surprisingly, they observe that mutation of the Phe-Gly (FG) repeat responsible for CPSF6 binding to the incoming HIV-1 capsid abrogates CPSF6 punctum formation. Perhaps more interestingly, they show that the removal of other domains of CPSF6, including the mixed-charge domain (MCD), does not affect the formation of HIV-1-induced CPSF6 puncta. The authors also present data suggesting that CPSF6 puncta form individual before fusing with nuclear speckles (NSs) and that the fusion of CPSF6 puncta to NSs requires the intrinsically disordered region (IDR) of the NS component SRRM2. While the study presents some interesting findings, there are some technical issues that need to be addressed and the amount of new information is somewhat limited. Also, the authors' finding that deletion of the CPSF6 MCD does not affect the formation of HIV-1-induced CPSF6 puncta contradicts recent findings of Jang et al. (https://doi.org/10.1093/nar/gkae769).

Comments on revisions:

The authors have generally addressed my comments.

---

## [Author Response]

The following is the authors’ response to the original reviews.

**Reviewer #1 (Public review):**
In recent years, our understanding of the nuclear steps of the HIV-1 life cycle has made significant advances. It has emerged that HIV-1 completes reverse transcription in the nucleus and that the host factor CPSF6 forms condensates around the viral capsid. The precise function of these CPSF6 condensates is under investigation, but it is clear that the HIV-1 capsid protein is required for their formation. This study by Tomasini et al. investigates the genesis of the CPSF6 condensates induced by HIV-1 capsid, what other co-factors may be required, and their relationship with nuclear speckels (NS). The authors show that disruption of the condensates by the drug PF74, added post-nuclear entry, blocks HIV-1 infection, which supports their functional role. They generated CPSF6 KO THP-1 cell lines, in which they expressed exogenous CPSF6 constructs to map by microscopy and pull down assays of the regions critical for the formation of condensates. This approach revealed that the LCR region of CPSF6 is required for capsid binding but not for condensates whereas the FG region is essential for both. Using SON and SRRM2 as markers of NS, the authors show that CPSF6 condensates precede their merging with NS but that depletion of SRRM2, or SRRM2 lacking the IDR domain, delays the genesis of condensates, which are also smaller.The study is interesting and well conducted and defines some characteristics of the CPSF6-HIV-1 condensates. Their results on the NS are valuable. The data presented are convincing.I have two main concerns. Firstly, the functional outcome of the various protein mutants and KOs is not evaluated. Although Figure 1 shows that disruption of the CPSF6 puncta by PF74 impairs HIV-1 infection, it is not clear if HIV-1 infection is at all affected by expression of the mutant CPSF6 forms (and SRRM2 mutants) or KO/KD of the various host factors. The cell lines are available, so it should be possible to measure HIV-1 infection and reverse transcription. Secondly, the authors have not assessed if the effects observed on the NS impact HIV-1 gene expression, which would be interesting to know given that NS are sites of highly active gene transcription. With the reagents at hand, it should be possible to investigate this too.

We thank the reviewer for her/his valuable feedback on our manuscript. We are pleased to see her/his appreciation of our results, and we did our utmost to address the highlighted points to further improve our work.

To correctly perform the infectivity assay, we generated stable cell clones—a process that required considerable time, particularly during the selection of clones expressing protein levels comparable to wild-type (WT) cells. To accurately measure infectivity, it was essential to use stable clones expressing the most important deletion mutant, ∆FG CPSF6, at levels similar to those of CPSF6 in WT cells (new Fig.5 A-B). Importantly, we assessed the reproducibility of our experiments by freezing and thawing these clones.

Regarding SRRM2, in THP-1 cells we were only able to achieve a knockdown, which still retains residual SRRM2 protein, albeit at much lower levels. Due to the essential role of SRRM2 in cell survival, obtaining a complete knockout in this cell line is not feasible, making it difficult to draw definitive conclusions from these experiments.

In contrast, 293T cells carrying the endogenous SRRM2 deletion mutant (ΔIDR) cannot be infected with replication-competent HIV-1, as they lack expression of CD4 and either CCR4 or CCR5. These cells were instead used to monitor the dynamics of CPSF6 puncta assembly within nuclear speckles. However, they are not a suitable model for studying the impact of the depletion of SRRM2 in viral infection.

Thus, we performed infectivity assays in a more relevant cell line for HIV-1 infection, THP-1 macrophage-like cells, using both a single-round virus and a replication-competent virus. The new results, shown in Figure 5 C-D, indicate that complete depletion of CPSF6 reduces infectivity, as measured by luciferase expression in a single-round infection (KO: ~65%; ΔFG: ~74%; compared to WT: 100% on average). Notably, a more pronounced defect in viral particle production was observed when WT virus was used for infection (KO: ~21%; ΔFG: ~16%; compared to WT: 100% on average). These findings support the referee’s insightful suggestion that the absence of CPSF6 could also impair HIV-1 gene expression.

**Reviewer #2 (Public review):**
Summary:HIV-1 infection induces CPSF6 aggregates in the nucleus that contain the viral protein CA. The study of the functions and composition of these nuclear aggregates have raised considerable interest in the field, and they have emerged as sites in which reverse transcription is completed and in the proximity of which viral DNA becomes integrated. In this work, the authors have mutated several regions of the CPSF6 protein to identify the domains important for nuclear aggregation, in addition to the alreadyknown FG region; they have characterized the kinetics of fusion between CPSF6 aggregates and SC35 nuclear speckles and have determined the role of two nuclear speckle components in this process (SRRM2, SUN2).Strengths:The work examines systematically the domains of CPSF6 of importance for nuclear aggregate formation in an elegant manner in which these mutants complement an otherwise CPSF6-KO cell line. In addition, this work evidences a novel role for the protein SRRM2 in HIV-induced aggregate formation, overall advancing our comprehension of the components required for their formation and regulation.Weaknesses:Some of the results presented in this manuscript, in particular the kinetics of fusion between CPSF6aggregates and SC35 speckles have been published before (PMID: 32665593; 32997983).The observations of the different effects of CPSF6 mutants, as well as SRRM2/SUN2 silencing experiments are not complemented by infection data which would have linked morphological changes in nuclear aggregates to function during viral infection. More importantly, these functional data could have helped stratify otherwise similar morphological appearances in CPSF6 aggregates.Overall, the results could be presented in a more concise and ordered manner to help focus the attention of the reader on the most important issues. Most of the figures extend to 3-4 different pages and some information could be clearly either aggregated or moved to supplementary data.

First, we thank the reviewer for her/his appreciation of our study and to give to us the opportunity to better explain our results and to improve our manuscript. We appreciate the reviewer’s positive feedback on our study, and we will do our best to address her/his concerns. In the meantime, we would like to clarify the focus of our study. Our research does not aim to demonstrate an association between CPSF6 condensates we use the term "condensates" rather than "aggregates," as aggregates are generally non-dynamic (Alberti & Hyman, 2021; Banani et al., 2017; Scoca et al., JMCB 2022), and our work specifically examines the dynamic behavior of CPSF6 puncta formed during infection and nuclear speckles. The association between CPSF6 puncta and NS has already been established in previous studies, as noted in the manuscript (PMID: 32665593; 32997983). The previous studies (PMID: 32665593; 32997983) showed that CPSF6 puncta colocalize with SC35 upon HIV infection and in the submitted study we study their kinetics.

About the point highlighted by the reviewer: "Kinetics of fusion between CPSF6-aggregates and SC35 speckles have been published before."

Our study differs from prior work PMID 32665593 because we utilize a full-length HIV genome, and we did not follow the integrase (IN) fluorescence in trans and its association with CPSF6 but we specifically assess if CPSF6 clusters form in the nucleus independently of NS factors and next to fuse with them. In the current study we evaluated the dynamics of formation of CPSF6/NS puncta, which it has not been explored before. Given this focus, we believe that our work offers a novel perspective on the molecular interactions that facilitate HIV / CPSF6-NS fusion.

We calculated that 27% of CPSF6 clusters were independent from NS at 6 h post-infection, compared to only 9% at 30 h. This likely reflects a reduction in individual clusters as more become fused with nuclear speckles over time. At the same time, these data suggest that the fusion process can begin even earlier. Indeed, it has been reported that in macrophages, the peak of viral nuclear import occurs before 6 h post-infection (doi: 10.1038/s41564-020-0735-8).

In addition, we have incorporated new experiments assessing viral infectivity in the absence of CPSF6, or in CPSF6-knockout cells expressing either a CPSF6 mutant lacking the FG peptide or the WT protein. As shown in our new Figure 5, these results demonstrate that the FG peptide is critical for viral replication in THP-1 cells.

For better clarity, we would like to specify that our study focuses on the role of SON, a scaffold factor of nuclear speckles, rather than SUN2 (SUN domain-containing protein 2), which is a component of the LINC (Linker of Nucleoskeleton and Cytoskeleton) complex.

As suggested by the reviewer, we have revised the text and combined figures to improve clarity and facilitate reader comprehension. We appreciate the constructive comment of the reviewer.

**Reviewer #3 (Public review):**
In this study, the authors investigate the requirements for the formation of CPSF6 puncta induced by HIV-1 under a high multiplicity of infection conditions. Not surprisingly, they observe that mutation of the Phe-Gly (FG) repeat responsible for CPSF6 binding to the incoming HIV-1 capsid abrogates CPSF6 punctum formation. Perhaps more interestingly, they show that the removal of other domains of CPSF6, including the mixed-charge domain (MCD), does not affect the formation of HIV-1-induced CPSF6 puncta. The authors also present data suggesting that CPSF6 puncta form individual before fusing with nuclear speckles (NSs) and that the fusion of CPSF6 puncta to NSs requires the intrinsically disordered region (IDR) of the NS component SRRM2. While the study presents some interesting findings, there are some technical issues that need to be addressed and the amount of new information is somewhat limited. Also, the authors' finding that deletion of the CPSF6 MCD does not affect the formation of HIV-1-induced CPSF6 puncta contradicts recent findings of Jang et al. (doi.org/10.1093/nar/gkae769).

We thank the reviewer for her/his thoughtful feedback and the opportunity to elaborate on why our findings provide a distinct perspective compared to those of Jang et al. (doi.org/10.1093/nar/gkae769).

One potential reason for the differences between our findings and those of Jang et al. could be the choice of experimental systems. Jang et al. conducted their study in HEK293T cells with CPSF6 knockouts, as described in Sowd et al., 2016 (doi.org/10.1073/pnas.1524213113). In contrast, our work focused on macrophage-like THP-1 cells, which share closer characteristics with HIV-1’s natural target cells.

Our approach utilized a complete CPSF6 knockout in THP-1 cells, enabling us to reintroduce untagged versions of CPSF6, such as wild-type and deletion mutants, to avoid potential artifacts from tagging. Jang et al. employed HA-tagged CPSF6 constructs, which may lead to subtle differences in experimental outcomes due to the presence of the tag.

Finally, our investigation into the IDR of SRRM2 relied on CRISPR-PAINT to generate targeted deletions directly in the endogenous gene (Lester et al., 2021, DOI: 10.1016/j.neuron.2021.03.026). This approach provided a native context for studying SRRM2’s role.

We will incorporate these clarifications into the discussion section of the revised manuscript.

**Reviewer #1 (Recommendations for the authors):**
(1) Figure 2E: The statistical analysis should be extended to the comparison between the "+HIV" samples.

We showed the statistics between only HIV+ cells now new Fig. 2D.

(2) Figure 4A top panel is out of focus.

We modified the figure now figure 6A.

**Reviewer #2 (Recommendations for the authors):**
(1) Some of the sentences could be rewritten for the sake of simplicity, also taking care to avoid overstatement.

We modified the sentences as best as we could.

(2) For instance: There is no evidence that "viral genomes in nuclear niches may be contributing to the formation of viral reservoirs" (lines 33-35).

We changed the sentence as follows: “Despite antiretroviral treatment, viral genomes can persist in these nuclear niches and reactivate upon treatment interruption, raising the possibility that they could play a role in the establishment of viral reservoirs.”

(3) Line 53: unclear sentence. "The initial stages of the viral life cycle have been understood....." The authors certainly mean reverse transcription, but as formulated this is not clear. The authors should also bear in mind that reverse transcription starts already in budding/just released virions.

We clarified the concept as follows: “the initial stages of the viral life cycle, such as the reverse transcription (the conversion of the viral RNA in DNA) and the uncoating (loss of the capsid), have been understood to mainly occur within the host cytoplasm.”

(4) Line 124: the results in Figure 1 are not at all explained in the text. PF74 does not act on CPSF6, it acts on CA and this in turn leads to CPS6 puncta disappearance.

PF74 binds the same hydrophobic pocket of the viral core as CPSF6. However, when viral cores are located within CPSF6 puncta, treatment with a high dose of PF74 leads to a rapid disassembly of these puncta, while viral cores remain detectable up to 2 hours post-treatment (Ay et al., EMBO J. 2024). Here, we simply describe what we observed by confocal microscopy. Said that HIV-Induced CPSF6 Puncta include both CPSF6 proteins and viral cores as we have now specified.

(5) Line 130; 'hinges into two key ...' should be 'hinges on'.

Thanks we modified it.

(6) Supplementary Figures are not cited sequentially in the text.

We have now modified the numbers of the supplementary figures according to their appearance in the text.

(7) Line 44: define FG.

We defined it.

**Reviewer #3 (Recommendations for the authors):**
Specific comments that the authors should address are outlined below.(1) As mentioned in the summary above, the authors' findings seem to be in direct contradiction with recent work published by Alan Engelman's lab in NAR. The authors should address the possible reason(s) for this discrepancy.

We mention the potential reasons for the differences in the results between our study and Engelman’s lab study in the discussion.

(2) The major finding here that deletion of the CFSF6 FG repeat prevents the formation of CFSP6 puncta is unsurprising, as the FG repeat is responsible for capsid binding. This has been reported previously and such mutants have been used as controls in other studies.

Our study demonstrates that the FG domain is the sole region responsible for the formation of CPSF6 puncta, rather than the LCR or MCD domains. The unique role of the FG domain in CPSF6 that promotes the formation of CPSF6 puncta without the help of the other IDRs during viral infection is a finding particularly novel, as it has not yet been reported in the literature.

(3) Line 339, the authors state: "incoming viral RNA has been observed to be sequestered in nuclear niches in cells treated with the reversible reverse transcriptase inhibitor, NEV. When macrophage-like cells are infected in the presence of NEV, the incoming viral RNA is held within the nucleus (Rensen et al., 2021; Scoca et al., 2023). This scenario is comparable to what is observed in patients undergoing antiretroviral therapy". In what way is this comparable to what is observed in individuals on ART? I see no basis for this statement. Sequestration of viral RNA in the nucleus is not the basis for maintaining the viral reservoir in individuals on therapy.

Thanks, we rephrased the sentence.

(4) General comment: analyzing single-cell-derived KO clones is very risky because of random clonal variability between individual cells in the population. If single-cell-derived clones are used, phenotypes could be confirmed with multiple, independent clones.

We used a clone completely KO for CPSF6 mainly to investigate the role of a specific domain in condensate formation and it will be difficult that clone selection could have introduced artifacts in this context. Other available clones retain residual endogenous protein, which prevents us from accurately assessing CPSF6 cluster formation in the various deletion mutants. A complete CPSF6 knockout is essential for studying puncta formation, as it eliminates potential artifacts arising from protein tags that could alter the phase separation properties of the protein under investigation.

(5) Line 214. "It is predicted to form two short α helices and a ß strand, arranged as: α helix - FG - ß strand - α helix". What is this based on? No citation is provided and no data are shown.

In fact, the statement "It is predicted to form two short α helices and a ß strand, arranged as: α helix - FG - ß strand - α helix" is based on the data shown in Figure 4E presenting data generated by PSIPRED.

(6) Figure 1B. "Luciferase values were normalized by total proteins revealed with the Bradford kit". What does this mean? I couldn't find anything explaining how the viral inputs were normalized.

The amount of the virus used is the same for all samples, we used MOI 10 as described in the legend of Figure 1. It is important to normalize the RLU (luciferase assay) with the total amount of proteins to be sure that we are comparing similar number of cells. Obviously, the cells were plated on the same amount on each well, the normalization in our case it is just an additional important control.

(7) I can't interpret what is being shown in the movies.

We updated the movie 1B and rephrased the movie legends and we added a new suppl. Fig.4B.

(8) Figure 5B. The differences seen are very small and of questionable significance. The data suggest that by 6 hpi, around 75% of HIV-induced CPSF6 puncta are already fused with NSs.

We calculated that 27% of CPSF6 clusters were independent from NS at 6 h post-infection, compared to only 9% at 30 h. This likely reflects a reduction in individual clusters as more become fused with nuclear speckles over time. At the same time, these data suggest that the fusion process can begin even earlier. Indeed, it has been reported that in macrophages, the peak of viral nuclear import occurs before 6 h post-infection (doi: 10.1038/s41564-020-0735-8).

(9) Figure 6. Immunofluorescence is not a good method for quantifying KD efficiency. The authors should perform western blotting to measure KD efficiency. This is an important point, because the effect sizes are small, quite likely due to incomplete KD.

We performed WB and quantified the results, which correlated with the IF data and their imaging analysis. These new findings have been incorporated into Figure 8A. Of note, deletion of the IDR of SRRM2 does not affect the number of SON puncta (Fig.8C), but significantly reduces the number of CPSF6 puncta in infected cells compared to those expressing full-length SRRM2 (Fig.8D).

(10) There are a variety of issues with the text that should be corrected.The authors use "RT" to mean both the enzyme (reverse transcriptase) and the process (reverse transcription). This is incorrect and will confuse the reader. RT refers to the enzyme (noun, not verb).The commonly used abbreviation for nevirapine is NVP, not NEV.In line 60, it is stated that the capsid contains 250 hexamers. This number is variable, depending on the size and shape of the capsid. By contrast, the capsid has exactly 12 pentamers.Line 75. Typo: "nuclear niches containing, such as like".Line 82. Typo: "the mechanism behinds".Line 102. Typo: "we aim to elucidate how these HIV-induced CPSF6 form".Line 107. Type: "CPSF6 is responsible for tracking the viral core" ("trafficking the viral core"?).

Thanks, we corrected all of them.